# Exact Computation of Any-Order Shapley Interactions for Graph Neural Networks

**Maximilian Muschalik**[1,*,✉]**, Fabian Fumagalli**[2,*]**, Paolo Frazzetto**[3]**, Janine Strotherm**[2]**,
Luca Hermes**[2]**, Alessandro Sperduti**[3,4,5]**, Eyke Hüllermeier**[1]**, Barbara Hammer**[2]

[1] LMU Munich, MCML    [2] Bielefeld University, CITEC    [3] University of Padova
[4] University of Trento    [5] Augmented Intelligence Center, FBK
[*] Equal Contribution    ✉ Corresponding Author: `maximilian.muschalik@lmu.de`

## ABSTRACT

Albeit the ubiquitous use of Graph Neural Networks (GNNs) in machine learning (ML) prediction tasks involving graph-structured data, their interpretability remains challenging. In explainable artificial intelligence (XAI), the Shapley Value (SV) is the predominant method to quantify contributions of individual features to a ML model's output. Addressing the limitations of SVs in complex prediction models, Shapley Interactions (SIs) extend the SV to groups of features. In this work, we explain single graph predictions of GNNs with SIs that quantify node contributions and interactions among multiple nodes. By exploiting the GNN architecture, we show that the structure of interactions in node embeddings are preserved for graph prediction. As a result, the exponential complexity of SIs depends only on the receptive fields, i.e. the message-passing ranges determined by the connectivity of the graph and the number of convolutional layers. Based on our theoretical results, we introduce GraphSHAP-IQ, an efficient approach to compute any-order SIs exactly. GraphSHAP-IQ is applicable to popular message-passing techniques in conjunction with a linear global pooling and output layer. We showcase that GraphSHAP-IQ substantially reduces the exponential complexity of computing exact SIs on multiple benchmark datasets. Beyond exact computation, we evaluate GraphSHAP-IQ's approximation of SIs on popular GNN architectures and compare with existing baselines. Lastly, we visualize SIs of real-world water distribution networks and molecule structures using a SI-Graph.

## 1 INTRODUCTION

Graph-structured data appears in many domains and real-world applications (Newman, 2018), such as molecular chemistry (Gilmer et al., 2017), water distribution networks (WDNs) (Ashraf et al., 2023), sociology (Borgatti et al., 2009), physics (Sanchez-Gonzalez et al., 2020), or human resources (Frazzetto et al., 2023). To leverage such structure in machine learning (ML) models, Graph Neural Networks (GNNs) emerged as the leading family of architectures that specifically exploit the graph topology (Scarselli et al., 2009). A major drawback of GNNs is the opacity of their predictive mechanism, which they share with most deep-learning based architectures (Amara et al., 2022). Reliable explanations for their predictions are crucial when model decisions have significant consequences (Zhang et al., 2024) or lead to new discoveries (McCloskey et al., 2019).
In explainable artificial intelligence (XAI), the Shapley Value (SV) (Shapley, 1953) is a prominent concept to assign contributions to entities of black box ML models (Lundberg & Lee, 2017; Covert et al., 2021; Chen et al., 2023). Entities typically represent features, data points (Ghorbani & Zou, 2019) or graph structures (Yuan et al., 2021; Ye et al., 2023). Although SVs yield an axiomatic attribution scheme, they do not give any insights into joint contributions of entities, known as *interactions*. Yet, interactions are crucial to understanding decisions of complex black box ML models (Wright et al., 2016; Sundararajan et al., 2020; Kumar et al., 2021). Shapley Interactions (SIs) (Grabisch & Roubens, 1999; Bordt & von Luxburg, 2023) extend the SV to include joint contributions of multiple entities. SIs satisfy similar axioms while providing interactions up to a maximum number of entities, referred to as the *explanation order*. In this context, SVs are the least complex SIs,

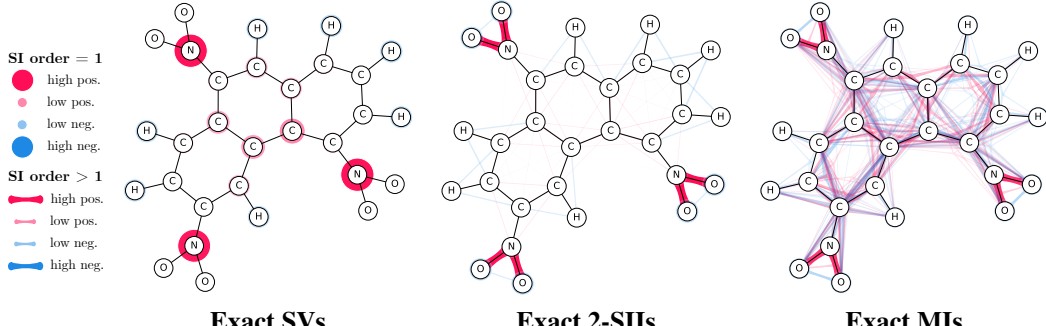

Figure 1: SI-Graphs overlayed on a molecule graph showing exact SIs for a molecule with 30 atoms from *MTG*. A GNN correctly identifies it as mutagenic. The SIs, in line with ground-truth knowledge, highlight the $NO_2$ groups. Computing exact SIs requires $2^{30} \approx 10^9$ model calls. GraphSHAP-IQ needs 7 693.

whereas Möbius Interactions (MIs) (or Möbius transform) (Harsanyi, 1963; Rota, 1964) are the most complex SIs by assigning contributions to every group of entities. Thus, SIs convey an adjustable explanation with an *accuracy-complexity trade-off* for interpretability (Bordt & von Luxburg, 2023). SVs, SIs and MIs are limited by exponential complexity, e.g. with 20 features already $2^{20} \approx 10^6$ model calls per explained instance are required. Consequently, practitioners rely on model-agnostic approximation methods (Lundberg & Lee, 2017; Fumagalli et al., 2023) or model-specific methods (Lundberg et al., 2020; Muschalik et al., 2024b) that exploit knowledge about the model's structure to reduce complexity. As a remedy for GNNs, the SV was applied as a heuristic on subgraphs (Ying et al., 2019; Ye et al., 2023), or approximated (Duval & Malliaros, 2021; Bui et al., 2024).

In this work, we address limitations of the SV for GNN explainability by computing the SIs visualized as the SI-Graph in Figure 1. Our method yields exact SIs by including GNN-specific knowledge and exploiting properties of the SIs. In contrast to existing methods (Yuan et al., 2021; Ye et al., 2023), we evaluate the GNN on node level without the need to cluster nodes into subgraphs. Instead of model-agnostic approximation (Duval & Malliaros, 2021; Bui et al., 2024), we provide structure-aware computation for graph prediction tasks, and prove that MIs of node embeddings indeed transfer to graph prediction for linear readouts. In summary, our approach is a model-specific computation of SIs for GNNs, akin to TreeSHAP (Lundberg et al., 2020) for tree-based models.

**Contribution.**    Our main contributions include:

(1) We introduce SIs among nodes and the SI-Graph for graph predictions of GNNs that address limitations of the SV and exploit graph and GNN structure with our theoretical results.

(2) We present GraphSHAP Interaction Quantification (GraphSHAP-IQ), an efficient method to compute exact any-order SIs in GNNs. For restricted settings requiring approximation, we extend GraphSHAP-IQ and propose several interaction-informed baseline methods.

(3) We show substantially reduced complexity when applying GraphSHAP-IQ on real-world benchmark datasets, and analyze SI-Graphs of a WDN and molecule structures.

(4) We find that interactions in deep readout GNNs are not restricted to the receptive fields.

**Related Work.**    SIs, enriching the SV (Shapley, 1953) with higher-order interactions, were introduced in game theory (Grabisch & Roubens, 1999), and modified for local interpretability in ML (Lundberg et al., 2020; Bordt & von Luxburg, 2023). The exponential complexity of the SV and SIs requires approximation by model-agnostic Monte Carlo sampling (Chen et al., 2023; Fumagalli et al., 2023) or by exploiting the data structure (Chen et al., 2019). Exact computation is only feasible with knowledge about the structure of the model and model-specific methods, such as TreeSHAP (Lundberg et al., 2020; Muschalik et al., 2024b), which is applicable to tree ensembles. Here, we present a model-specific method applicable to GNNs and graph prediction tasks, which computes exact SIs by exploiting the graph and GNN structure.

For local explanations of GNNs and graph prediction tasks, a variety of perturbation-based methods (Ying et al., 2019; Luo et al., 2020; Yuan et al., 2021) have been proposed, which output isolated

subgraphs, whereas Pope et al. (2019) introduce node attributions. In this work, we use perturbations via node maskings and output attributions for all nodes, and all subset of nodes up to the given explanation order, which additively decomposes the graph prediction of the GNN.

In context of GNN interpretability, the SV was applied in GraphSVX (Duval & Malliaros, 2021) on single nodes with a structure-aware approximation for node predictions. For graph prediction tasks, however, GraphSVX proposes a model-agnostic approximation. SubgraphX (Yuan et al., 2021) and SAME (Ye et al., 2023) use the SV to assess the quality of isolated subgraphs. Recently, model-agnostic approximation of pairwise SIs have shown to improve isolated subgraph detections (Bui et al., 2024). In contrast to existing work, we propose a structure-aware method that efficiently computes exact SVs for single nodes and exact any-order SIs for subset of nodes for graph prediction tasks of GNNs. Our explanation is based on all possible interactions (MIs) and additively decomposes the prediction. For a more detailed discussion of related work, we refer to Appendix C.

## 2 BACKGROUND

In Section 2.1, we introduce SIs that provide an adjustable *accuracy-complexity* trade-off for explanations (Bordt & von Luxburg, 2023). In this context, SVs are the simplest and MIs the most complex SIs. In Section 2.2, we introduce GNNs, whose structure we exploit in Section 3 to efficiently compute any-order SIs. A summary of notations can be found in Appendix A.

### 2.1 EXPLANATION COMPLEXITY: FROM SHAPLEY VALUES TO MÖBIUS INTERACTIONS

Concepts from cooperative game theory, such as the SV (Shapley, 1953), are prominent in XAI to interpret predictions of a black box ML model via feature attributions (Strumbelj & Kononenko, 2014; Lundberg & Lee, 2017). Formally, a cooperative game $\nu : \mathcal{P}(N) \to \mathbb{R}$ is defined, where individual features $N = \{1, \dots, n\}$ act as players and achieve a payout for every group of players in the power set $\mathcal{P}(N)$. To obtain feature attributions for the prediction of a single instance, $\nu$ typically refers to the model's prediction given only a subset of feature values. Since classical ML models cannot handle missing feature values, different methods have been proposed, such as model retraining (Strumbelj et al., 2009), conditional expectations (Lundberg & Lee, 2017; Aas et al., 2021; Frye et al., 2021), marginal expectations (Janzing et al., 2020) and baseline imputations (Lundberg & Lee, 2017; Sundararajan et al., 2020). In high-dimensional feature spaces, retraining models or approximating feature distributions is infeasible, imputing absent features with a baseline, known as Baseline Shapley (BShap) (Sundararajan & Najmi, 2020), is the prevalent method (Lundberg & Lee, 2017; Sundararajan et al., 2017; Jethani et al., 2022). We now first introduce the MIs as a backbone of additive contribution measures. Later in Section 3, we exploit sparsity of MIs for GNNs to compute the SV and any-order SIs.

**Möbius Interactions (MIs)** $m : \mathcal{P}(N) \to \mathbb{R}$ are a fundamental concept of cooperative game theory and provide the basis for different summary measures. The MI is

$$m(S) := \sum_{T \subseteq S} (-1)^{|S|-|T|} \nu(T) \text{ and they recover } \nu(T) = \sum_{S \subseteq T} m(S) \text{ for all } S, T \subseteq N. \quad (1)$$

From the MIs, every game value can be additively recovered, and MIs are the unique measure with this property (Harsanyi, 1963; Rota, 1964). The MI of a subset $S \subseteq N$ can thus be interpreted as the *pure additive contribution* that is exclusively achieved by a coalition of all players in $S$, and cannot be attributed to any subgroup of $S$. The MIs are further a basis of the vector space of games (Grabisch, 2016), and therefore every measure of contribution, such as the SV or the SIs, can be directly recovered from the MIs, cf. Appendix E.3.

**Shapley Values (SVs)** for players $i \in N$ of a cooperative game $\nu$ are the weighted average

$$\phi^{\text{SV}}(i) := \sum_{T \subseteq N \setminus i} \frac{1}{n \cdot \binom{n-1}{|T|}} \Delta_i(T) \text{ with } \Delta_i(T) := \nu(T \cup i) - \nu(T)$$

over marginal contributions $\Delta_i(T)$. It was shown (Shapley, 1953) that the SV is the unique attribution method that satisfies desirable axioms: *linearity* (the SV of linear combinations of games, e.g., model ensembles, coincides with the linear combinations of the individual SVs), *dummy* (features that do not change the model's prediction receive zero SV), *symmetry* (if a model does not change

its prediction when switching two features, then both receive the same SV), and lastly *efficiency* (the sum of all SVs equals the difference between the model's prediction $\nu(N)$ and the featureless prediction $\nu(\emptyset)$). We may normalize $\nu$, such that $\nu(\emptyset) = 0$, which does not affect the SVs. The SV assigns attributions to individual features, which distribute the MIs that contain feature $i$, cf. Appendix E.3. However, the SV does not provide any insights about *feature interactions*, i.e. the joint contribution of multiple features to the prediction. Yet, in practice, understanding complex models requires investigating interactions (Slack et al., 2020; Sundararajan et al., 2020; Kumar et al., 2021). While the SVs are limited in their expressivity, the MIs are difficult to interpret due to the exponential number of components. The SIs provide a framework to bridge both concepts.

**Shapley Interactions (SIs)** explore model predictions beyond individual feature attributions, and provide additive contribution for all subsets up to *explanation order* $k = 1, \ldots, n$. More formally, the SIs $\Phi_k : \mathcal{P}_k(N) \to \mathbb{R}$ assign interactions to subsets of $N$ up to size $k$, summarized in $\mathcal{P}_k(N)$. The SIs decompose the model's prediction with $\nu(N) = \sum_{S \subseteq N, |S| \leq k} \Phi_k(S)$. The least complex SIs are the SVs, which are obtained with $k = 1$. For $k = n$, the SIs are the MIs with $2^n$ components, which provide the most faithful explanation of the game but entail the highest complexity. SIs are constructed based on extensions of the marginal contributions $\Delta_i(T)$, known as discrete derivatives (Grabisch & Roubens, 1999). For two players $i, j \in N$, the discrete derivative $\Delta_{ij}(T)$ for a subset $T \subseteq N \setminus ij$ is defined as $\Delta_{ij}(T) := \nu(T \cup ij) - \nu(T) - \Delta_i(T) - \Delta_j(T)$, i.e., the joint contribution of adding both players together minus their individual contributions in the presence of $T$. This recursion is extended to any subset $S \subseteq N$ and $T \subseteq N \setminus S$. A positive value of the discrete derivative $\Delta_S(T)$ indicates synergistic effects, a negative value indicates redundancy, and a value close to zero indicates no joint information of all players in $S$ given $T$. The Shapley Interaction Index (SII) (Grabisch & Roubens, 1999) provides an axiomatic extension of the SV and summarizes the discrete derivatives in the presence of all possible subsets $T$ as

$$\phi^{\text{SII}}(S) = \sum_{T \subseteq N \setminus S} \frac{1}{(n - |S| + 1) \cdot \binom{n-|S|}{|T|}} \Delta_S(T) \quad \text{with} \quad \Delta_S(T) := \sum_{L \subseteq S} (-1)^{|S|-|L|} \nu(T \cup L).$$

Given an explanation order $k$, the $k$-Shapley Values ($k$-SIIs) (Lundberg et al., 2020; Bordt & von Luxburg, 2023) construct SIs recursively based on the SII, such that the interactions of SII and $k$-SII for the highest order coincide. Alternatively, the Shapley Taylor Interaction Index (STII) (Sundararajan & Najmi, 2020) and the Faithful Shapley Interaction Index (FSII) (Tsai et al., 2023) have been proposed, cf. Appendix E.2. In summary, SIs provide a flexible framework of increasingly complex and faithful contributions ranging from the SV ($k = 1$) to the MIs ($k = n$). Given the MIs, it is possible to reconstruct SIs of arbitrary order, cf. Appendix E.3. In Section 3, we will exploit the sparse structure of MIs of GNNs to efficiently compute any-order SIs.

## 2.2 GRAPH NEURAL NETWORKS

GNNs are neural networks specifically designed to process graph input (Scarselli et al., 2009). A graph $g = (V, E, \mathbf{X})$ consists of sets of nodes $V = \{v_1, ..., v_n\}$, edges $E \subset V \times V$ and $d_0$-dimensional node features $\mathbf{X} = [\mathbf{x}_1, ..., \mathbf{x}_n]^t \in \mathbb{R}^{n \times d_0}$, where $\mathbf{x}_i$ are the node features of node $v_i \in V$. A *message passing* GNN leverages the structural information of the graph $g$ to iteratively aggregate node feature information of a given node $v \in V$ within its *neighborhood* $\mathcal{N}(v) := \{u \in V \mid e_{uv} \in E\}$. More precisely, in each iteration $k \in \{1, ..., \ell\}$, the $d_k$-dimensional $k$-th hidden node features $\mathbf{H}^{(k)} = [\mathbf{h}_1^{(k)}, \ldots, \mathbf{h}_n^{(k)}]^t \in \mathbb{R}^{n \times d_k}$ are computed node-wise by

$$\mathbf{h}_i^{(0)} := \mathbf{x}_i, \quad \mathbf{h}_i^{(k)} := \rho^{(k)}(\mathbf{h}_i^{(k-1)}, \psi(\{\{\varphi^{(k)}(\mathbf{h}_j^{(k-1)}) \mid u_j \in \mathcal{N}(v_i)\}\})), \quad (2)$$

where $\{\{\cdot\}\}$ indicates a multiset and $\rho^{(k)}$ and $\varphi^{(k)}$ are arbitrary (aggregation) functions acting on the corresponding spaces. Moreover, $\psi$ is implemented as a permutation-invariant function, ensuring independence of both the order and number of neighboring nodes, and allows for an embedding of multisets as vectors. The node embedding function is thus $f_i(\mathbf{X}) := \mathbf{h}_i^{(\ell)}$. For *graph prediction* tasks, the representations of the last nodes $\mathbf{H}^{(\ell)}$ must be aggregated in a fixed-size graph embedding for the downstream task. More formally, this is achieved with an additional permutation-invariant pooling function $\Psi$ and a parametrized output layer $\sigma : \mathbb{R}^{d_\ell} \to \mathbb{R}^{d_{\text{out}}}$, where $d_{\text{out}}$ corresponds to the output dimension. The output of a GNN for graph-level inference is defined as

$$f_g(\mathbf{X}) := \sigma\big(\Psi(\{\{f_i(\mathbf{X}) \mid v_i \in V\}\})\big). \quad (3)$$

For *graph classification*, class probabilities are obtained from $f_g(\mathbf{X})$ through *softmax* activation.

## 3   ANY-ORDER SHAPLEY INTERACTIONS FOR GRAPH NEURAL NETWORKS

In the following, we are interested in explaining the prediction of a GNN $f_g$ for a graph $g$ with respect to nodes. We aim to decompose a model's prediction into SIs $\Phi_k$ visualized by a SI-Graph.

**Definition 3.1** (SI-Graph). *The SI-Graph is an undirected hypergraph $g_k^{SI} := (N, \mathcal{P}_k(N), \Phi_k)$ with node attributes $\Phi_k(i)$ for $i \in N$ and hyperedge attributes $\Phi_k(S)$ for $2 \leq |S| \leq k$.*

The simplest SI-Graph displays the SVs ($k = 1$) as node attributes, whereas the most complex SI-Graph displays the MIs ($k = n$) as node and hyperedge attributes, illustrated in Figure 1. The complexity of the SI-Graph is adjustable by the explanation order $k$, which determines the maximum hyperedge order. The sum of all contributions in the SI-Graph yields the model's prediction (for regression) or the model's logits for the predicted class (for classification). This choice is natural for an additive contribution measure due to additivity in the logit-space. To compute SIs, we introduce the GNN-induced graph game $\nu_g$ with a node masking strategy in Section 3.1. The graph game is defined on all nodes and describes the output given a subset of nodes, where the remaining are masked. Computing SIs on the graph game defines a perturbation-based and a decomposition-based GNN explanation (Yuan et al., 2023), which is an extension of node attributions (Agarwal et al., 2023). In Section 3.2, we show that GNNs with linear global pooling and output layer satisfy an invariance property for the node game associated with the node embeddings (Theorem 3.3). This invariance implies sparse MIs for the graph game (Proposition 3.6), which determines the complexity of MIs by the corresponding receptive fields (Theorem 3.7), which substantially reduces the complexity of SIs in our experiments. In Section 3.3, we introduce GraphSHAP-IQ, an efficient algorithm to exactly compute and estimate SIs on GNNs. All proofs are deferred to Appendix B.

### 3.1   A COOPERATIVE GAME FOR SHAPLEY INTERACTIONS ON GRAPH NEURAL NETWORKS

Given a GNN $f_g$, we propose the graph game for which we compute *axiomatic and fair* SIs.

**Definition 3.2** (GNN-induced Graph and Node Game). *For a graph $g = (V, E, \mathbf{X})$ and a GNN $f_g$, we let $N := \{i : v_i \in V\}$ be the node indices and define the graph game $\nu_g : \mathcal{P}(N) \to \mathbb{R}$ as*

$$\nu_g(T) := f_{g,\hat{y}}(\mathbf{X}^{(T)}) \text{ with } \mathbf{X}^{(T)} := (\mathbf{x}_1^{(T)}, ..., \mathbf{x}_n^{(T)})^t \in \mathbb{R}^{n \times d_0} \text{ and } \mathbf{x}_i^{(T)} := \begin{cases} \mathbf{x}_i & \text{if } i \in T, \\ \mathbf{b} & \text{if } i \notin T, \end{cases}$$

*with $i \in N$ and baseline $\mathbf{b} \in \mathbb{R}^{d_0}$. In graph regression $f_{g,\hat{y}} \equiv f_g$ and for graph classification $f_{g,\hat{y}}$ is the component of the predicted class $\hat{y}$ of $f_g$. We further introduce the (multi-dimensional) node game $\nu_i : \mathcal{P}(N) \to \mathbb{R}^{d_\ell}$ as $\nu_i(T) := f_i(\mathbf{X}^{(T)})$ for $i \in N$ and each node $v_i \in V$.*

The graph game outputs the prediction of the GNN for a subset of nodes $T \subseteq N$ by masking all node features of nodes $v_i$ with $i \in N \setminus T$ using a suitable baseline $\mathbf{b}$, illustrated in Figure 2, left. Computing such SVs is known as BShap (Sundararajan & Najmi, 2020) and a prominent approach for feature attributions (Lundberg & Lee, 2017; Covert et al., 2021; Chen et al., 2023). As a baseline $\mathbf{b}$, we propose the average of each node feature over the whole graph. By definition, the prediction of the GNN is given by $\nu_g(N) = f_g(\mathbf{X})$, and due to the efficiency axiom, the sum of contributions in the SI-Graph yields the model's prediction, and thus a decomposition-based GNN explanation (Yuan et al., 2023). The graph and the node game are directly linked by Equation (3) as

$$\nu_g(T) = f_{g,\hat{y}}(\mathbf{X}^{(T)}) = \sigma_{\hat{y}}(\Psi(\{\{f_i(\mathbf{X}^{(T)})\} \mid v \in V\})) = \sigma_{\hat{y}}(\Psi(\{\{\nu_i(T)\} \mid i \in N\})), \quad (4)$$

where $\sigma_{\hat{y}}$ outputs the component of $\sigma$ for the predicted class $\hat{y}$. The number of convolutional layers $\ell$ determines the *receptive field*, i.e. the message-passing range defined by its $\ell$-hop neighborhood

$$\mathcal{N}_i^{(\ell)} := \{j \in N \mid d_g(i, j) \leq \ell\} \text{ with } d_g(i, j) := \text{length of shortest path from } v_j \text{ to } v_i \text{ in } g.$$

Consequently, the node game $\nu_i$ is unaffected by maskings outside its $\ell$-hop neighborhood.

**Theorem 3.3** (Node Game Invariance). *For a graph $g$ and an $\ell$-Layer GNN $f_g$, let $\nu_i$ be the GNN-induced node game with $i \in N$. Then, $\nu_i$ satisfies the invariance $\nu_i(T) = \nu_i(T \cap \mathcal{N}_i^{(\ell)})$ for $T \subseteq N$.*

**Node Masking:** Computing SIs on the graph game is a perturbation-based explanation (Yuan et al., 2023), where also other masking strategies were proposed (Agarwal et al., 2023); for example, node masks (Ying et al., 2019; Yuan et al., 2021), edge masks (Luo et al., 2020; Schlichtkrull et al., 2021) or node feature masks (Agarwal et al., 2023). Our method is not limited to a specific masking strategy as long as it defines an invariant game (Theorem 3.3). We implement our method with the well-established and theoretically understood BShap (Sundararajan & Najmi, 2020). Alternatively, the $T$-induced subgraph could be used, but GNNs are fit to specific graph topologies, such as molecules, and perform poorly on isolated subgraphs (Alsentzer et al., 2020). Note that, different masking strategies may emphasize different aspects of GNNs, which is important future work. Due to the invariance, we show that MIs and SIs of the graph game are sparse. To obtain our theoretical results, we require a structural assumption.

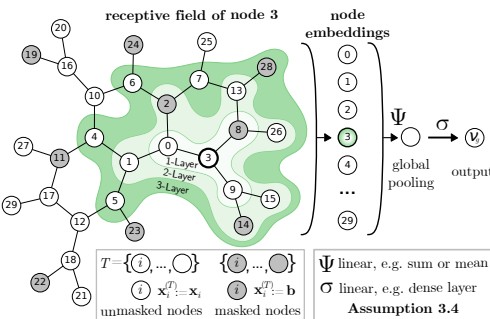

Figure 2: Illustration of the graph game $\nu_g$. Masked nodes (grey) are imputed by baseline $\mathbf{b}$, and embeddings are determined by the receptive field (left). Subsequently, a linear pooling ($\Psi$) and output layer ($\sigma$) yield the GNN-induced graph game output.

**Assumption 3.4** (GNN Architecture). *We require the global pooling $\Psi$ and the output layer $\sigma$ to be linear functions, e.g. $\Psi$ is a mean or sum pooling operation and $\sigma$ is a dense layer.*

**Linearity Assumption:** In our experiments, we show that popular GNN architectures yield competitive performances under Assumption 3.4 on multiple benchmark datasets. In fact, such an assumption should not be seen as a hindrance, as it is the norm in GNN benchmark evaluations (Errica et al., 2020). Furthermore, simple global pooling functions, such as *sum* or *mean*, are adopted in many GNN architectures (Xu et al., 2019; Wu et al., 2022), while more sophisticated pooling layers do not always translate into empirical benefits (Mesquita et al., 2020; Grattarola et al., 2021). Likewise, a linear output layer is a common design choice, and the advantage of deeper output layers must be validated for each task (You et al., 2020).

### 3.2 COMPUTING EXACT SHAPLEY AND MÖBIUS INTERACTIONS FOR THE GRAPH GAME

Given a GNN-induced graph game $\nu_g$ from Definition 3.2 with Assumption 3.4, i.e. $\Psi$ and $\sigma$ are linear, then the MIs of each node game are restricted to the $\ell$-hop neighborhood. Intuitively, maskings outside the receptive field do not affect the node embedding. Consequently, we show that the MIs of the graph game are restricted by all existing $\ell$-hop neighborhoods. More formally, due to the invariance of the node games (Theorem 3.3), the MIs for subsets that are not fully contained in the $\ell$-hop neighborhood $\mathcal{N}_i^{(\ell)}$ are zero.

**Lemma 3.5** (Trivial Node Game Interactions). *Let $m_i : \mathcal{P}(N) \to \mathbb{R}^{d_\ell}$ be the MIs of the GNN-induced node game $\nu_i$ for $i \in N$ under Assumption 3.4. Then, $m_i(S) = \mathbf{0}$ for all $S \not\subseteq \mathcal{N}_i^{(\ell)}$.*

Lemma 3.5 yields that the node game interactions outside of the $\ell$-hop neighborhood do not have to be computed. Due to Assumption 3.4, the interactions of the GNN-induced graph game are equally zero for subsets that are not fully contained in any $\ell$-hop neighborhood.

**Proposition 3.6** (Trivial Graph Game Interactions). *Let $m_g : \mathcal{P}(N) \to \mathbb{R}$ be the MIs of the GNN-induced graph game $\nu_g$ under Assumption 3.4 and let $\mathcal{I} := \bigcup_{i \in N} \mathcal{P}(\mathcal{N}_i^{(\ell)})$ be the set of non-trivial interactions. Then, $m_g(S) = 0$ for all $S \subseteq N$ with $S \notin \mathcal{I}$.*

$\mathcal{I}$ is the set of non-trivial MIs, whose size depends on the receptive field of the GNN. The size of $\mathcal{I}$ also directly determines the required model calls to compute SIs.

**Theorem 3.7** (Complexity). *For a graph $g$ and an $\ell$-Layer GNN $f_g$, computing MIs and SIs on the GNN-induced graph game $\nu_g$ requires $|\mathcal{I}|$ model calls. The complexity is thus bounded by*

$$|\mathcal{I}| \leq \sum_{i \in N} 2^{|\mathcal{N}_i^{(\ell)}|} \leq n \cdot 2^{n_{\max}^{(\ell)}} \leq n \cdot 2^{\frac{d_{\max}^{\ell+1} - 1}{d_{\max} - 1}},$$

*where $n_{\max}^{(\ell)} := \max_{i \in N} |\mathcal{N}_i^{(\ell)}|$ is the size of the largest $\ell$-hop neighborhood and $d_{\max}$ is the maximum degree of the graph instance.*

In other words, Theorem 3.7 shows that the complexity of MIs (originally $2^n$) for GNNs depends at most *linearly on the size* of the graph $n$. Moreover, the complexity depends exponentially on the connectivity $d_{\max}$ of the graph instance and the number of convolutional layers $\ell$ of the GNN. Note that this is a very rough theoretical bound. In our experiments, we empirically demonstrate that in practice for many instances exact SIs can be computed, even for large graphs ($n > 100$). Besides this upper bound, we empirically show that the *graph density*, which is the ratio of edges compared to the number of edges in a fully connected graph, is an efficient proxy for the complexity.

### 3.3 GRAPHSHAP-IQ: AN EFFICIENT ALGORITHM FOR SHAPLEY INTERACTIONS

Building on Theorem 3.7, we propose GraphSHAP-IQ, an efficient algorithm to compute SIs for GNNs. At the core is the exact computation of SIs, outlined in Algorithm 1, which we then extend for approximation in restricted settings. Moreover, we propose interaction-informed baseline methods that directly exclude zero-valued SIs, which, however, still require all model calls for exact computation. To compute exact SIs, GraphSHAP-IQ identifies the set of non-zero MIs $\mathcal{I}$ based on the given graph instance (line 1). The GNN is then evaluated for all maskings contained in $\mathcal{I}$ (line 2). Given these GNN predictions, the MIs for all interactions in $\mathcal{I}$ are computed (line 3). Based on the computed MIs, the SIs are computed using the conversion formulas

---

**Algorithm 1** GraphSHAP-IQ

**Require:** Graph $g = (V, E, \mathbf{X})$, $\ell$-Layer GNN $f_g$, SI order $k$.

1: $\mathcal{I} \leftarrow \bigcup_{i \in N} \mathcal{P}(\mathcal{N}_i^{(\ell)})$
2: $\nu_g \leftarrow [f_g(\mathbf{X}^{(T)})]_{T \in \mathcal{I}}$
3: $m_g \leftarrow [\texttt{MI}(\nu_g, S)]_{S \in \mathcal{I}}$
4: $\Phi_k \leftarrow \texttt{MItoSI}(m_g, k)$
5: **return** MIs $m_g$, SIs $\Phi_k$

---

(line 4). Lastly, GraphSHAP-IQ outputs the exact MIs and SIs. In restricted settings, computing exact SIs could still remain infeasible. We thus propose an approximation variant of GraphSHAP-IQ by introducing a hyperparameter $\lambda$, which limits the highest order of computed MIs in line 1. Hence, GraphSHAP-IQ outputs exact SIs, if $\lambda = n_{\max}^{(\ell)}$, thereby requiring the optimal budget. For a detailed description of GraphSHAP-IQ and the interaction-informed variants, we refer to Appendix D.

## 4 EXPERIMENTS

In this section, we empirically evaluate GraphSHAP-IQ for GNN explainability, and showcase a substantial reduction in complexity for exact SIs (Section 4.1), benefits of approximation (Section 4.2), and explore the SI-Graph for WDNs and molecule structures (Section 4.3). Following Amara et al. (2022), we trained a Graph Convolutional Network (GCN) (Kipf & Welling, 2017), Graph Isomorphism Network (GIN) (Xu et al., 2019), and Graph Attention Network (GAT) (Velickovic et al., 2018) on eight real-world chemical datasets for graph classification and a WDN for graph regression, cf. Table 1. All models adhere to Assumption 3.4 and report comparable test accuracies (Errica et al., 2020; You et al., 2020). All experiments are based on shapiq (Muschalik et al., 2024a) and details can be found in Appendix F or at https://github.com/FFmgll/GraphSHAP-IQ.

Table 1: Summary of datasets and model accuracy.

| Dataset | *Dataset Description* | | | | *Model Accuracy by Layer (%)* | | | | | | | | | Speed-Up | | |
| | Graphs | $d_{out}$ | Nodes (avg) | Density (avg) | GCN | | | GAT | | | GIN | | | | | |
| | | | | | 1 | 2 | 3 | 1 | 2 | 3 | 1 | 2 | 3 | 1 | 2 | 3 |
|---|---|---|---|---|---|---|---|---|---|---|---|---|---|---|---|---|
| Benzene (*BNZ*) (Sánchez-Lengeling et al., 2020) | 12000 | 2 | 20.6 | 22.8 | 84.2 | 88.6 | 90.4 | 83.1 | 85.1 | 85.7 | 84.9 | 90.5 | 90.8 | $10^4$ | $10^3$ | $10^2$ |
| FluorideCarbonyl (*FLC*) (Sánchez-Lengeling et al., 2020) | 8671 | 2 | 21.4 | 21.6 | 82.4 | 83.9 | 83.9 | 82.4 | 82.2 | 82.4 | 84.6 | 87.2 | 87.1 | $10^4$ | $10^3$ | $10^2$ |
| Mutagenicity (*MTG*) (Kazius et al., 2005) | 4337 | 2 | 30.3 | 18.3 | 77.8 | 80.7 | 80.3 | 72.6 | 73.6 | 74.8 | 77.8 | 77.4 | 77.5 | $10^5$ | $10^4$ | $10^2$ |
| AlkaneCarbonyl (*ALC*) (Sánchez-Lengeling et al., 2020) | 1125 | 2 | 21.4 | 21.5 | 98.7 | 97.8 | 99.1 | 98.2 | 96.3 | 97.3 | 96.9 | 97.3 | 97.8 | $10^4$ | $10^3$ | $10^2$ |
| PROTEINS (*PRT*) (Borgwardt et al., 2005) | 1113 | 2 | 39.1 | 42.4 | 75.2 | 71.1 | 74.0 | 75.3 | 60.5 | 79.8 | 79.3 | 74.9 | 67.7 | $10^5$ | $10^3$ | $10^2$ |
| ENZYMES (*ENZ*) (Borgwardt et al., 2005) | 600 | 6 | 32.6 | 32.0 | 34.2 | 37.5 | 35.8 | 32.5 | 35.0 | 35.8 | 36.7 | 35.0 | 39.2 | $10^6$ | $10^4$ | $10^3$ |
| COX2 (*CX2*) (Sutherland et al., 2003) | 467 | 2 | 41.2 | 10.6 | 87.2 | 86.1 | 87.2 | 81.9 | 87.2 | 85.1 | 84.0 | 85.1 | 85.1 | $10^9$ | $10^8$ | $10^6$ |
| BZR (*BZR*) (Sutherland et al., 2003) | 405 | 2 | 35.8 | 13.0 | 90.1 | 87.7 | 90.2 | 88.9 | 86.4 | 87.7 | 88.9 | 88.9 | 88.9 | $10^8$ | $10^6$ | $10^4$ |

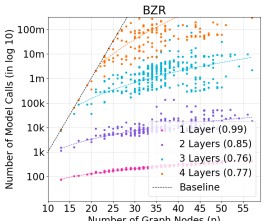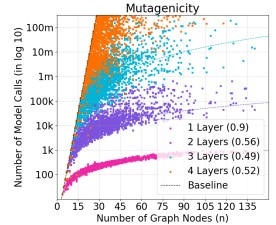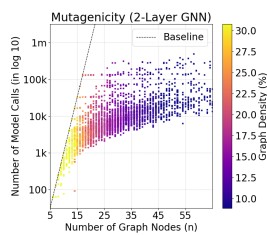

Figure 3: Complexity of GraphSHAP-IQ in model calls (in $\log 10$) by number of nodes for all graphs of *BZR* (left) and *MTG* (middle, right) visualized by number of convolutional layers (left, middle) and graph density GNN (right). While model-agnostic baselines scale exponentially (dashed lines), GrahpSHAP-IQ scales approximately linearly with graph sizes ($R^2$ of log-curves in braces).

## 4.1 COMPLEXITY ANALYSIS OF GRAPHSHAP-IQ FOR EXACT SHAPLEY INTERACTIONS

In this experiment, we empirically validate the benefit of exploiting graph and GNN structures to compute *exact SIs* with GraphSHAP-IQ. The complexity is measured by the number of evaluations of the GNN-induced graph game, i.e. the number of model calls of the GNN, which is the limiting factor of SIs, cf. runtime analysis in Appendix G.2. For every graph in the benchmark datasets, described in Table 1, we compute the complexity of GraphSHAP-IQ, where the first upper bound from Theorem 3.7 is used if $\max_{i \in N} |\mathcal{N}_i^{(\ell)}| > 23$, i.e. the complexity exceeds $2^{23} \approx 8.3 \times 10^6$. Figure 3 displays the log-scale complexity (y-axis) by the number of nodes $n$ (x-axis) for *BZR* (left) and *MTG* (middle, right) for varying number of convolutional layers $\ell$ (left, middle) and by graph density for a 2-Layer GNN (right). The model-agnostic baseline is represented by a dashed line. For results on all datasets, see Appendix G.1. Figure 3 shows that the computation of SIs is substantially reduced by GraphSHAP-IQ. Even for large graphs with more than $100$ nodes, where the baseline requires over $10^{30}$ model calls, many instances can be exactly computed for 1-Layer and 2-Layer GNNs with fewer than $10^5$ evaluations. In fact, the complexity grows *linearly* with graph size across the dataset, as shown by high $R^2$ scores of fitted logarithmic curves. Figure 3 (right) shows that the graph density is an efficient proxy of complexity, with higher values for instances near the baseline.

## 4.2 INTERACTION-INFORMED APPROXIMATION OF SHAPLEY INTERACTIONS

For densely connected graphs and GNNs with many layers, exact computation of SIs might still be infeasible. We, thus, evaluate the *approximation of SIs* with GraphSHAP-IQ, current state-of-the-art model-agnostic baselines (implemented in shapiq), and our proposed interaction-informed variants. For the SV (order 1), we apply *KernelSHAP* (Lundberg & Lee, 2017), *Unbiased KernelSHAP* (Covert & Lee, 2021), *k-additive SHAP* (Pelegrina et al., 2023), *Permutation Sampling* (Castro et al., 2009), *SVARM* (Kolpaczki et al., 2024a), and *L-Shapley* (Chen et al., 2019). We estimate $k$-SII (order 2 and 3) with *KernelSHAP-IQ* (Fumagalli et al., 2024), *Inconsistent KernelSHAP-IQ* (Fumagalli et al., 2024), *Permutation Sampling* (Tsai et al., 2023), *SHAP-IQ* (Fumagalli et al., 2023), and *SVARM-IQ* (Kolpaczki et al., 2024b). For each baseline, we use the interaction-informed variant, cf. Appendix D.3. We select graphs containing $20 \leq n \leq 40$ nodes for the *MTG*, *PRT*, and *BZR* benchmark datasets. For each graph instance, we compute ground-truth SIs via GraphSHAP-IQ and evaluate all methods using the same number of model calls, which is the main driver of runtime, cf. Appendix G.2. Figure 4 (left) displays the average MSE (lower is better) for varying the model calls. GraphSHAP-IQ outperforms the baselines in settings with a majority of lower-order MIs. Figure 4 (middle) compares average runtime and MSE for varying explanation orders at GraphSHAP-IQ's ground-truth budget. Notably, GraphSHAP-IQ is among the fastest methods, and remains unaffected by increasing explanation order. Moreover, for all baselines (except permutation sampling), the interaction-informed variants substantially improve the approximation quality and runtime. Consequently, noisy estimates of SIs are substantially improved (Figure 4, right).

## 4.3 REAL-WORLD APPLICATIONS OF SHAPLEY INTERACTIONS AND THE SI-GRAPH

We now apply GraphSHAP-IQ in real-world applications and include further results in Appendix G. **Monitoring water quality** in WDNs requires insights into a dynamic system governed by local

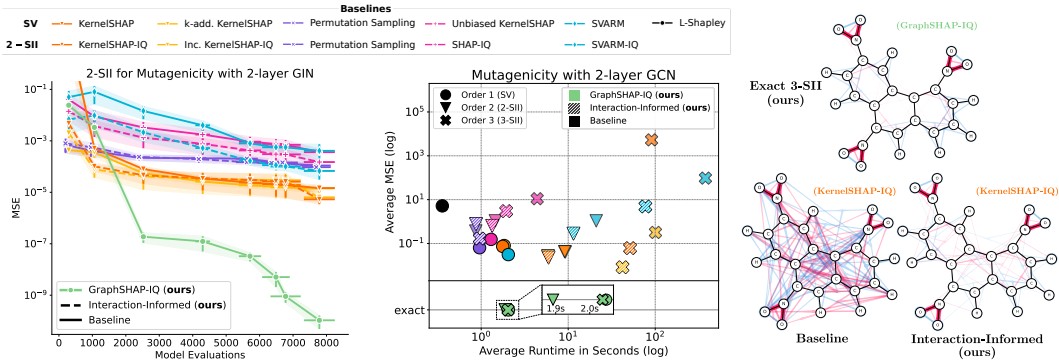

Figure 4: Approximation of SIs with GraphSHAP-IQ (green) and model-agnostic baselines for *MTG* (left). At budgets, where GraphSHAP-IQ reaches exact SIs, the baselines achieve varying estimation qualities and computational costs (middle) which leads to different estimated explanations, especially without interaction-informed baselines (right).

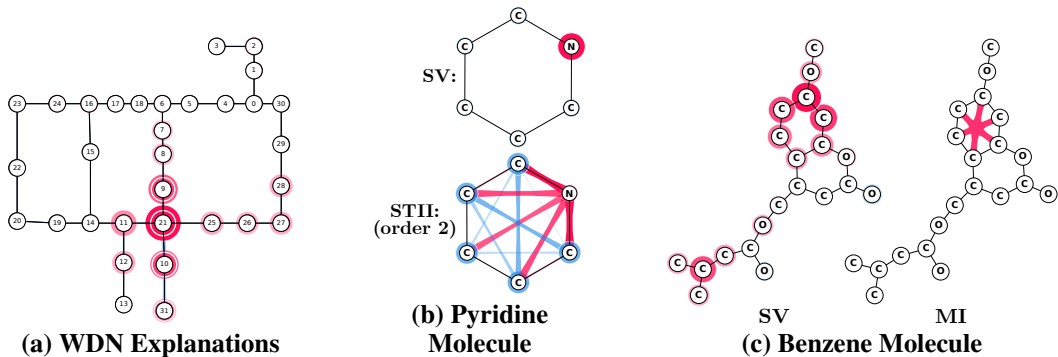

Figure 5: Exact SIs values for three example graph structures. SVs illustrate the trajectory of chlorination levels in a WDN (a). STII (order 2) values showcase that a Pyridine molecule is not classified as benzene (b), and the largest positive MI for a benzene molecule is the benzene ring (c).

partial differential equations. Here, we investigate the spread of chlorine as a graph-level regression of a WDN, where a GNN predicts the fraction of nodes chlorinated after some time. Based on the Hanoi WDS (Vrachimis et al., 2018), we create a temporal WaterQuality (*WAQ*) dataset containing 1 000 graphs consisting of 30 time steps. We train and explain a simple GNN, which processes node and edge features like chlorination level at each node and water flow between nodes. Figures 5 and 11 show that 2-SIIs spread over the WDS aligned with the water flow. Therein, mostly first-order interactions influence the time-varying chlorination levels.

**Benzene rings in molecules** are structures consisting of six carbon (C) atoms connected in a ring with alternating single and double bonds. We expect a well-trained GNN to identify benzene rings to incorporate higher-order MIs (order $\geq 6$). Figure 5 shows two molecules and their SI-Graphs computed by GraphSHAP-IQ. The Pyridine molecule in Figure 5 (b) is correctly predicted to be non-benzene as the hexagonal configuration features a nitrogen (N) instead of a carbon, which is confirmed by the SVs highlighting the nitrogen. STIIs of order 2 reveal that the MI of nitrogen is zero and interactions with neighboring carbons are non-zero, presumably due to higher-order MIs, since STII distributes all higher-order MIs to the pairwise STIIs. In addition, STIIs among the five carbon atoms impede the prediction towards the benzene class. Interestingly, opposite carbons coincide with the highest negative interaction. The MIs for a benzene molecule with 21 atoms in Figure 5 (c) reveal that the largest positive MI coincides with the 6-way MI of the benzene ring.

**Mutagenicity of molecules** is influenced by compounds like nitrogen dioxide ($NO_2$) (Kazius et al., 2005). Figure 1 shows SIs for a *MTG* molecule, which GNN identifies as mutagenic. 2-SIIs and MIs both show that not the nitrogen atom but the interactions of the $NO_2$ bonds contributed the most.

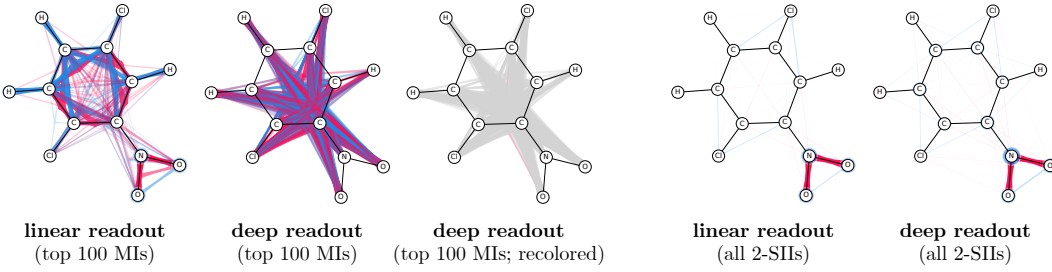

**linear readout**        **deep readout**         **deep readout**          **linear readout**        **deep readout**
(top 100 MIs)             (top 100 MIs)            (top 100 MIs; recolored)   (all 2-SIIs)              (all 2-SIIs)

Figure 6: MIs and 2-SII scores for a 2-layer GCN trained on *MTG* with a linear readout and a deep readout. Deep readout produces interactions outside the receptive field which are recolored in grey.

## 5 COMPARISON OF LINEAR AND DEEP READOUTS

Theorem 3.4 imposes the use of a linear readout, which is a limitation of our method, though it remains commonly used in practice (You et al., 2020). In this section, we compare SIs for GNNs with both linear and non-linear readouts. We train a 2-layer GCN architecture with both linear and non-linear (2-layer perceptron) readouts on *MTG*, where both models achieve comparable performance. Figure 6 shows that non-linear readouts produce substantially different MIs that are *not restricted* to the receptive fields. Lower-order explanations are similar, indicating the correct reasoning of both models ($NO_2$ group signalling mutagenicity). We conclude that linear readout restricts the interactions of the GNN to the graph structure and its receptive fields. In contrast, non-linear readouts enable interactions that extend beyond the receptive fields of the GNN. Many SV-based XAI methods for GNNs (Yuan et al., 2021; Zhang et al., 2022; Ye et al., 2023; Bui et al., 2024) implicitly rely on the assumption that interactions outside the receptive fields are negligible, which should be evaluated carefully for non-linear readouts.

## 6 LIMITATIONS AND FUTURE WORK

We presented GraphSHAP-IQ, an efficient method to compute SIs that applies to all popular message passing techniques in conjunction with a linear global pooling and output layer. **Assumption 3.4** is a common choice for GNNs (Errica et al., 2020; Xu et al., 2019; Wu et al., 2022) and does not necessarily yield lower performance (Mesquita et al., 2020; You et al., 2020; Grattarola et al., 2021), which is confirmed by our experiments. However, exploring non-linear choices that preserve trivial MIs is important for future research. **Masking node features with a fixed baseline**, known as BShap, preserves the topology of the graph structure and is a well-established approach (Sundararajan et al., 2020). Nevertheless, alternatives such as induced subgraphs, edge-removal, or learnable masks, could emphasize other properties of the GNN. Lastly, **approximation of SIs** with GraphSHAP-IQ and interaction-informed baselines substantially improved the estimation, where novel methods tailored to Proposition 3.6 are promising future work. Our results may further be applied to other models with spatially restricted features, such as convolutional neural networks.

## 7 CONCLUSION

We introduced the GNN-induced graph game, a cooperative game for GNNs on graph prediction tasks that outputs the model's prediction given a set of nodes. The remaining nodes are masked using a baseline for node features, corresponding to the well-established BShap (Sundararajan et al., 2020). We showed that under linearity assumptions on global pooling and output layers, the complexity of computing exact SVs, SIs, and MIs on any GNN is determined solely by the receptive fields. Based on our theoretical results, we presented GraphSHAP-IQ and interaction-informed variants of existing baselines, to efficiently compute any-order SIs for GNNs. We show that GraphSHAP-IQ and interaction-informed baselines substantially reduces the complexity of SIs on multiple real-world benchmark datasets and propose to visualize SIs as the SI-Graph. By computing the SI-Graph, we discover trajectories of chlorine in WDNs and important molecule substructures, such as benzene rings or $NO_2$ groups.

ETHICS STATEMENT

This paper presents work aiming to advance the field of ML and specifically the field of XAI. There are many potential societal consequences of our work. Our research holds significant potential for positive societal impact, particularly in areas like the natural sciences (e.g., chemistry and biology) and network analytics. Our work can positively impact ML adoption and potentially reveal biases or unwanted behavior in ML systems.

However, we recognize that the increased explainability provided by XAI also carries ethical risks. There is the potential for "explainability-based white-washing", where organizations, firms, or institutions might misuse XAI to justify questionable actions or outcomes. With responsible use, XAI can amplify the positive impacts of ML, ensuring its benefits are realized while minimizing harm.

REPRODUCIBILITY STATEMENT

The python code for GraphSHAP-IQ is available at https://github.com/FFmgll/GraphSHAP-IQ, and can be used on any *graph game*, a class specifically tailored to the shapiq package (Muschalik et al., 2024a). We include formal proofs of all claims made in the paper in Appendix B. We further describe the experimental setup and details regarding reproducibility in Appendix F. Our experimental results, setups and plots can be reproduced by running the corresponding scripts. The datasets and their sources are described in Table 3.

ACKNOWLEDGMENTS

We gratefully thank the anonymous reviewers for their valuable feedback for improving this work! We thank André Artelt for the EPyT-Flow toolbox support. Fabian Fumagalli and Maximilian Muschalik gratefully acknowledge funding by the Deutsche Forschungsgemeinschaft (DFG, German Research Foundation): TRR 318/1 2021 – 438445824. Janine Strotherm gratefully acknowledges funding from the European Research Council (ERC) under the ERC Synergy Grant WaterFutures (Grant agreement No. 951424). Luca Hermes and Paolo Frazzetto gratefully acknowledge funding by "SAIL: SustAInable Lifecycle of Intelligent Socio-Technical Systems," funded by the Ministry of Culture and Science of the State of North Rhine-Westphalia under grant NW21-059A. Additionally, Paolo Frazzetto gratefully acknowledges funding by Amajor SB S.p.A. Alessandro Sperduti gratefully acknowledges the support of the PNRR project FAIR - Future AI Research (PE00000013), Concession Decree No. 1555 of October 11, 2022, CUP C63C22000770006.

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

ORGANISATION OF THE SUPPLEMENT MATERIAL

# A  NOTATION

We use lower-case letters to represent the cardinalities of the subsets, e.g. $s := |S|$. A summary of notations is given in Table 2.

Table 2: Notation Table

| Notation | Description |
|---|---|
| **Data Notations** | |
| $g = (V, E, \mathbf{X})$ | Graph instance to explain, with nodes $V$, edges $E$, and node features $\mathbf{X}$ |
| $V := \{v_1, \ldots, v_n\}$ | Set of nodes in $g$ |
| $E \subset V \times V$ | Set of edges in $g$ |
| $d_0$ | Number of node features |
| $\mathbf{x_i} \in \mathbb{R}^{d_0}$ | Node feature vector of node $v_i$ |
| $\mathbf{X} \in \mathbb{R}^{n \times d_0}$ | Node feature matrix for all nodes in $g$ |
| $\mathbf{b} \in \mathbb{R}^{d_0}$ | Baseline node features used for masking |
| $N := \{i : v_i \in V\}$ | Set of graph node indices |
| $n := |N|$ | Number of nodes in the graph instance |
| $i \in N$ | Index of node $v_i \in V$ in $g$ |
| $\mathcal{N}(v) \subseteq N$ | Node indices of the neighborhood of node $v$ |
| **GNN Notations** | |
| $f_g(\mathbf{X})$ | GNN prediction for graph $g$ and node features $\mathbf{X}$ |
| $f_{g,\hat{y}}(\mathbf{X})$ | GNN prediction of class $\hat{y}$ for graph $g$ and node features $\mathbf{X}$ |
| $\ell$ | Number of graph convolutional layers |
| $k = 1, \ldots, \ell$ | Intermediate graph convolutional layers |
| $d_k$ | Number of features in node embedding at layer $k$ |
| $\mathbf{h}_i^{(k)} \in \mathbb{R}^{d_k}$ | Node embedding vector for node $v_i$ at layer $k$ |
| $\mathbf{H}^{(k)} := [\mathbf{h}_1^{(k)}, \ldots, \mathbf{h}_n^{(k)}]$ | Node embedding matrix of the graph instance at layer $k$ |
| $\Psi$ | (Linear) permutation-invariant global pooling function |
| $\sigma$ | (Linear) readout layer of the GNN |
| $\mathcal{N}_i^{(\ell)} \subseteq N$ | Node indices of the $\ell$-hop neighborhood of node $v_i$ |
| $n_{\max}^{(\ell)} := \max_{i \in N} |\mathcal{N}_i^{(\ell)}|$ | Size of the largest $\ell$-hop neighborhood of the graph instance |
| $d_{\max}$ | Maximum node degree of the graph instance |
| **Masking Notations** | |
| $T \subseteq N$ | Set of node indices, typically used for masked predictions, where $N \setminus T$ are masked |
| $\mathbf{x}_i^{(T)} := \begin{cases} \mathbf{x}_i & \text{if } i \in T, \\ \mathbf{b} & \text{if } i \notin T, \end{cases}$ | Masked node features of nodes $v_i$, where nods in $N \setminus T$ are masked with baseline $\mathbf{b}$ |
| $\mathbf{X}^{(T)} = [\mathbf{x}_1^{(T)}, \ldots, \mathbf{x}_n^{(T)}]$ with $T \subseteq N$ | Masked node feature matrix of $g$, where nodes in $N \setminus T$ are masked with $\mathbf{b}$ |
| $f_g(\mathbf{X}^{(T)})$ | Masked prediction of the GNN using masked node features $\mathbf{X}^{(T)}$ |
| $f_i(\mathbf{X}^{(T)})$ with $i \in N$ | Masked node embedding $f_i$ for node $v_i$ evaluated at masked node features $\mathbf{X}^{(T)}$ |
| **Game Theory Notations** | |
| $\mathcal{P}(N)$ | Power set of $N$, i.e. all possible sets of node indices used to define the game |
| $\mathcal{P}_k(N)$ | Power set of $N$ up to sets of size $k$, i.e. the sets for which the final SI explanation is computed |
| $\nu_g(T) := f_{g,\hat{y}}(\mathbf{X}^{(T)})$ | Graph game evaluated at $T \subseteq N$, i.e. GNN prediction $f_g$ of class $\hat{y}$ with masked node features $\mathbf{X}^{(T)}$ |
| $\nu_i(T) := f_i(\mathbf{X}^{(T)})$ for $i \in N$ | Graph node game evaluated at $T \subseteq N$, i.e. GNN embedding $f_i$ of node $i$ with masked node features $\mathbf{X}^{(T)}$ |
| $S \subseteq N$ | Set of node indices, used to refer to an interaction, i.e. the sets for which attributions are computed |
| $m_g(S)$ | MI for a set $S \subseteq N$ of the graph game $\nu_g$ |
| $m_i(S)$ | MI for a set $S \subseteq N$ of the graph node game $\nu_i$ |
| $\mathcal{I} := \bigcup_{i \in N} \mathcal{P}(\mathcal{N}_i^{(\ell)})$ | Set of non-trivial MIs, i.e. $m_g(S) = 0$, if $S \notin \mathcal{I}$ |
| $\Phi_k(S)$ | Final SI explanation of order $k$ evaluated at a set $S$ of size at most $k$ |

# B   PROOFS

## B.1   PROOF OF THEOREM 3.3

*Proof.* By definition of $\nu_i$, we need to show that $f_i(\mathbf{X}^{(T)}) = f_i(\mathbf{X}^{(T \cap \mathcal{N}_i^{(\ell)})})$ holds. We prove by induction over $\ell$. Denote $f_i^{(\ell)}$ the corresponding node embedding function for the $\ell$-layer GNN For a GNN with $\ell = 1$, it is immediately clear from Equation (2) that

$$f_i^{(1)}(\mathbf{X}^{(T)}) = \rho^{(1)}(\mathbf{x}_i^{(T)}, \psi(\{\{\varphi^{(1)}(\mathbf{x}_j^{(T)}) \mid u_j \in \mathcal{N}_i^{(1)}\}\})) = f_i^{(1)}(\mathbf{X}^{(T \cap \mathcal{N}_i^{(1)})}).$$

Next, assume $f_i^{(\ell-1)}(\mathbf{X}^{(T)}) = f_i^{(\ell-1)}\left(\mathbf{X}^{(T \cap \mathcal{N}_i^{(\ell-1)})}\right)$ holds for any GNN, $i \in N$ and $T \subseteq N$. Then for a GNN with $\ell$ layers, we have

$$\begin{aligned}
f_i^{(\ell)}(\mathbf{X}^{(T)}) &= \rho^{(\ell)}(f_i^{(\ell-1)}(\mathbf{X}^{(T)}), \psi(\{\{\varphi^{(\ell)}(f_j^{(\ell-1)}(\mathbf{X}^{(T)})) \mid v_j \in \mathcal{N}_i^{(1)}\}\})) \\
&= \rho^{(\ell)}(f_i^{(\ell-1)}(\mathbf{X}^{(T \cap \mathcal{N}_i^{(\ell-1)})}), \psi(\{\{\varphi^{(\ell)}(f_j^{(\ell-1)}(\mathbf{X}^{(T \cap \mathcal{N}_j^{(\ell-1)})})) \mid v_j \in \mathcal{N}_i^{(1)}\}\})) \\
&= \rho^{(\ell)}(f_i^{(\ell-1)}(\mathbf{X}^{(T \cap \mathcal{N}_i^{(\ell)})}), \psi(\{\{\varphi^{(\ell)}(f_j^{(\ell)}(\mathbf{X}^{(T \cap \mathcal{N}_i^{(\ell)})})) \mid v_j \in \mathcal{N}_i^{(1)}\}\})) \\
&= f_i^{(\ell)}(\mathbf{X}^{(T \cap \mathcal{N}_i^{(\ell)})})
\end{aligned}$$

where we have used that $\mathcal{N}_i^{(\ell-1)} \subseteq \mathcal{N}_i^{(\ell)}$ and $\mathcal{N}_j^{(\ell-1)} \subseteq \mathcal{N}_i^{(\ell)}$ for $j \in \mathcal{N}_i^{(1)}$ together with the invariance (induction hypothesis) of $f_i^{(\ell-1)}$ and $f_j^{(\ell-1)}$. $\square$

## B.2   PROOF OF LEMMA 3.5

*Proof.* Let $i \in N$ and $S \nsubseteq \mathcal{N}_i^{(\ell)}$. Our goal is to show that $m_i(S) = 0$. For every subset $T \subseteq N$, we can define disjoint sets

$$T_i^+ := T \cap \mathcal{N}_i^{(\ell)} \text{ and } T_i^- := T \cap (N \setminus \mathcal{N}_i^{(\ell)}), \text{ such that } T = T_i^+ \sqcup T_i^-,$$

with $T_i^+ \subseteq \mathcal{N}_i^{(\ell)}$ and $T_i^- \subseteq N \setminus \mathcal{N}_i^{(\ell)}$. For $S = S_i^+ \sqcup S_i^-$ the assumption $S \nsubseteq \mathcal{N}_i^{(\ell)}$ implies that $S_i^-$ is not empty, i.e. $s_i^- := |S_i^-| > 0$. Furthermore, due to the node game invariance, cf. Theorem 3.3, we have $\nu_i(T) = \nu_i(T_i^+)$ for every $T \subseteq N$. In the following, lower-case letters represent the corresponding cardinalities of the subsets, e.g. $t_i^+ := |T_i^+|$. For the MI of interest $S$, the sum, by definition of MIs, ranges over $\mathcal{P}(S)$. Thus, since $S_i^+, S_i^- \in \mathcal{P}(S)$, we have for every $T_i^+ \subseteq S_i^+$ and every $T_i^- \subseteq S_i^-$ that the union $T_i^+ \sqcup T_i^- \in \mathcal{P}(S)$. On the other hand, every $T \in \mathcal{P}(S)$ can be uniquely decomposed into $T = T_i^+ \sqcup T_i^-$. Hence, instead of summing over all $T \in \mathcal{P}(S)$, we may also sum over all subsets $T = T_i^+ \cup T_i^-$ with $T_i^+ \subseteq S_i^+$ and $T_i^- \subseteq S_i^-$. The MI for $S$ is then

$$\begin{aligned}
m_i(S) &= \sum_{T \subseteq S} (-1)^{s-t} \nu_i(T) \\
&= \sum_{T_i^+ \subseteq S_i^+} \sum_{T_i^- \subseteq S_i^-} (-1)^{s-(t_i^+ + t_i^-)} \nu_i\left(T_i^+ \sqcup T_i^-\right) \\
&= \sum_{T_i^+ \subseteq S_i^+} \sum_{T_i^- \subseteq S_i^-} (-1)^{s-(t_i^+ + t_i^-)} \nu_i\left(T_i^+\right) \\
&= \sum_{T_i^+ \subseteq S_i^+} (-1)^{s-t_i^+} \nu_i\left(T_i^+\right) \sum_{T_i^- \subseteq S_i^-} (-1)^{t_i^-} \\
&= \sum_{T_i^+ \subseteq S_i^+} (-1)^{s-t_i^+} \nu_i\left(T_i^+\right) \sum_{t_i^-=0}^{s_i^-} \binom{s_i^-}{t_i^-} (-1)^{t_i^-} \\
&= \sum_{T_i^+ \subseteq S_i^+} (-1)^{s-t_i^+} \nu_i\left(T_i^+\right) (1-1)^{s_i^-} = 0,
\end{aligned}$$

where we used the node game invariance in the third equation, the binomial theorem $(a+b)^n = \sum_{k=0}^n \binom{n}{k} a^{n-k} b^k$ in the second last equation, and $s_i^- > 0$ in the last equation. $\square$

### B.3 PROOF OF PROPOSITION 3.6

*Proof.* The proof will be based on two important properties of the MIs.

**Lemma B.1.** *The MI of a constant game $\nu \equiv c \in \mathbb{R}$ is*

$$m_c(S) = \begin{cases} c, & \text{if } S = \emptyset, \\ 0, & \text{otherwise.} \end{cases}$$

*Proof.* The MI for a constant game is computed for $S = \emptyset$ as $m(\emptyset) = \nu(\emptyset) = c$ and for $s := |S| > 0$ as

$$m(S) = \sum_{T \subseteq S} (-1)^{s-t} \nu(T) = c \sum_{T \subseteq S} (-1)^{s-t} = c \cdot (-1)^s \sum_{t=0}^{s} \binom{s}{t} (-1)^t = c \cdot (-1)^s (1-1)^s = 0,$$

where we have used the binomial theorem $(a+b)^n = \sum_{k=0}^{n} \binom{n}{k} a^{n-k} b^k$. $\qquad\square$

The second property is the linearity of the MI in terms of a linear combination of games.

**Lemma B.2** (Linearity (Fujimoto et al., 2006))**.** *For a linear combination of two games $\nu := c \cdot \nu_1 + \nu_2$ for a constant $c \in \mathbb{R}$, the MI of $\nu$ is given as*

$$m_\nu(S) = c \cdot m_{\nu_1}(S) + m_{\nu_2}(S).$$

*for all $S \subseteq N$.*

*Proof.* This result follows from Theorem 4.9 and Lemma 4.1 in Fujimoto et al. (2006). However, it may also be verified directly

$$m_\nu(S) = \sum_{T \subseteq S} (-1)^{s-t} \nu(T) = \sum_{T \subseteq S} (-1)^{s-t} (c \cdot \nu_1(T) + \nu_2(T)) = c \cdot m_{\nu_1}(T) + m_{\nu_2}(T).$$

$\qquad\square$

Next, let $S \notin \bigcup_{i \in N} \mathcal{P}(\mathcal{N}_i^{(\ell)})$. Our goal is to show that $m_g(S) = 0$. By Lemma 3.5, we have that $m_i(S) = \mathbf{0} \in \mathbb{R}^{d_\ell}$ for all $i \in N$ and node games $\nu_i : \mathcal{P}(N) \to \mathbb{R}^{d_\ell}$. We can thus define a new game $\nu_\Psi : \mathcal{P}(N) \to \mathbb{R}^{d_\ell}$ as $\nu_\Psi(T) := \Psi(\{\{\nu_i(T) \mid i \in N\}\})$ for $T \subseteq N$. Due to Assumption 3.4, $\Psi$ is linear (a linear combination of inputs), and by Lemma B.1, constant shifts do not affect the MIs. We therefore assume that $\Psi(\{\{\mathbf{0}\}, \dots \{\mathbf{0}\}\}) = \mathbf{0}$ for any number of zero vectors $\mathbf{0}$. By the linearity of the MI (Lemma B.2), we have then

$$m_\Psi(S) = \Psi(\{\{m_i(S) \mid i \in N\}\}) = \mathbf{0}.$$

Lastly, $\nu_g$ represents the logits of the predicted class in the GNN, i.e. $\nu_g(T) = \sigma(\nu_\Psi(T))$. The output layer $\sigma$ transforms the games defined as the components of the (multi-dimensional) output of $\nu_\Psi$ to an vector of size $d_{\text{out}}$, where $\sigma_{\hat{y}}$ returns the component used in the graph game $\nu_g$. Again, by Lemma B.1, constant shifts do not affect the MIs, and we let $\sigma_{\hat{y}}(\mathbf{0}) = 0$, to obtain

$$m_g(S) = \sigma_{\hat{y}}(m_\Psi(S)) = \sigma_{\hat{y}}(\mathbf{0}) = 0,$$

and likewise for graph regression, which concludes the proof. $\qquad\square$

### B.4 PROOF OF THEOREM 3.7

By Proposition 3.6, the set of non-trivial MIs is given by $\mathcal{I}$. To compute exact MIs it is therefore necessary to compute all MIs contained in $\mathcal{I}$. Clearly, to compute the MI $m_g(S)$ for any $S \subseteq N$, it is by definition of the MI necessary to evaluate $\nu_g(S)$. Hence, the complexity of computing exact MIs cannot be lower than $|\mathcal{I}|$. Given now all graph game evaluations $\nu_g(T)$ with $T \in \mathcal{I}$, we proceed to show that no additional evaluation is required. In fact, to compute the MI $m_g(S)$ for $S \in \mathcal{I}$, we require all game evaluations $\nu(T)$ with $T \subseteq S$. By definition of $\mathcal{I}$ (Proposition 3.6), there exists a node index $i \in N$, such that $S \in \mathcal{P}(\mathcal{N}_i^{(\ell)})$. Since $S \subseteq \mathcal{N}_i^{(\ell)}$ it follows immediately that all $T \subseteq S$ satisfy $T \in \mathcal{P}(\mathcal{N}_i^{(\ell)})$, and hence $T \in \mathcal{I}$, which we have already computed. This finishes

the proof that exact computation requires $|\mathcal{I}|$ graph game evaluations and hence GNN model calls. Additionally, the number of elements in $\mathcal{I}$ is trivially bounded by the number of subsets in $N$, i.e. $2^n$ and further

$$|\mathcal{I}| = \left| \bigcup_{i \in N} \mathcal{P}(\mathcal{N}_i^{(\ell)}) \right| \leq \sum_{i \in N} |\mathcal{P}(\mathcal{N}_i^{(\ell)})| = \sum_{i \in N} 2^{|\mathcal{N}_i^{(\ell)}|} \leq n \cdot 2^{n_{\max}^{(\ell)}},$$

where each $\ell$-hop neighborhood was bounded by $n_{\max}^{(\ell)} := \max_{i \in N} |\mathcal{N}_i^{(\ell)}|$. Lastly, we bound the number of nodes in the $\ell$-hop neighborhood by bounding the number of nodes in each hop with the maximum degree $d_{\max}$ as

$$n_{\max}^{(\ell)} \leq 1 + d_{\max} + d_{\max}^2 + \cdots + d_{\max}^\ell = \frac{d_{\max}^{\ell+1} - 1}{d_{\max} - 1},$$

where we have used the formula for geometric progression. With this bound we obtain the final bound

$$|\mathcal{I}| \leq n \cdot 2^{n_{\max}^{(\ell)}} \leq n \cdot 2^{\frac{d_{\max}^{\ell+1} - 1}{d_{\max} - 1}},$$

which concludes the proof.

## C   Detailed Comparison of GraphSHAP-IQ and Related Work

The following contains a detailed summary of the related and relevant work. We further discuss distinctions of GraphSHAP-IQ with specific methods from related work in Appendix C.1. Moreover, we establish a connection between GraphSHAP-IQ and L-Shapley in Appendix C.2

The SV (Shapley, 1953) was applied in XAI for local (Lundberg & Lee, 2017; Covert et al., 2021) and global (Casalicchio et al., 2019; Covert et al., 2020) model interpretability, or data valuation (Ghorbani & Zou, 2019). The Myerson value (Myerson, 1977) is a variant of the SV for games restricted to graph components. Extensions of the SV to interactions for ML were proposed by Lundberg et al. (2020); Sundararajan et al. (2020); Tsai et al. (2023); Bordt & von Luxburg (2023) Due to the exponential complexity, model-specific methods for tree-based models for SVs (Lundberg et al., 2020; Yu et al., 2022), any-order SIs (Zern et al., 2023; Muschalik et al., 2024b) and MIs (Hiabu et al., 2023) were proposed. Model-agnostic approximation methods cover the SV (Lundberg & Lee, 2017; Chen et al., 2019; Kolpaczki et al., 2024a), SIs (Fumagalli et al., 2023; Kolpaczki et al., 2024b; Fumagalli et al., 2024), and MIs (Kang et al., 2024).
Instance-wise explanations on GNNs were proposed via perturbations (Ying et al., 2019; Luo et al., 2020; Yuan et al., 2021; Schlichtkrull et al., 2021), gradients (Pope et al., 2019; Sánchez-Lengeling et al., 2020; Schnake et al., 2022; Xiong et al., 2023) or surrogate models (Vu & Thai, 2020; Huang et al., 2023). GNN explanations are given in terms of nodes (Pope et al., 2019; Huang et al., 2023) and subgraphs (Ying et al., 2019; Luo et al., 2020; Yuan et al., 2021). Other use paths (Schnake et al., 2022; Huang et al., 2023) and edges (Schlichtkrull et al., 2021). For perturbation-based attributions, maskings for nodes (Ying et al., 2019; Yuan et al., 2021) or edges (Luo et al., 2020; Schlichtkrull et al., 2021) were introduced. The SV was applied in perturbation-based methods to assess the quality of subgraphs (Yuan et al., 2021; Zhang et al., 2022; Ye et al., 2023), approximate SVs (Duval & Malliaros, 2021; Akkas & Azad, 2024) or for pre-defined motifs (Perotti et al., 2022). Recently, pairwise SIs were approximated to discover subgraph explanations (Bui et al., 2024).
In contrast to existing work, we compute exact SVs on node level. Moreover, we exploit graph and GNN structure to compute SIs that uncover complex interactions, formally prove that for linear readouts interaction structures of node embeddings are preserved for graph predictions.

### C.1   Detailed Differences between GraphSHAP-IQ and Related Methods

In this section, we compare GraphSHAP-IQ with TreeSHAP (Lundberg et al., 2020), TreeSHAP-IQ (Muschalik et al., 2024b), GraphSVX (Duval & Malliaros, 2021), SubgraphX (Yuan et al., 2021), GStarX (Zhang et al., 2022) SAME (Ye et al., 2023), and recent work by Bui et al. (2024).

**TreeSHAP (Lundberg et al., 2020)**   TreeSHAP is a model-specific computation method to explain predictions of decision trees and ensembles of decision trees. TreeSHAP defines a cooperative game based on perturbations via the conditional distribution learned by the tree (path-dependent TreeSHAP) or via interventions using a background dataset (interventional TreeSHAP). Given these game definitions, TreeSHAP efficiently computes exact SVs in polynomial time by exploiting the tree structure. Recently, extensions of TreeSHAP for interventional perturbations (Zern et al., 2023), and with TreeSHAP-IQ for path-dependent perturbations (Muschalik et al., 2024b) were proposed to efficiently compute exact any-order SIs in polynomial time. Similar to GraphSHAP-IQ, TreeSHAP and TreeSHAP-IQ are model-specific variants for efficient and exact computation of SVs and SIs. However, TreeSHAP and TreeSHAP-IQ are only applicable to tree-based models, whereas GraphSHAP-IQ is only applicable to GNNs and other model classes with spatially-restricted features. While both methods are model-specific variants to efficiently compute SIs, these methods are applied on fundamentally different model classes and exhibit completely different computation schemes.

**Myerson-Taylor Interactions (Bui et al., 2024)**   This work highlights the opportunities and significance of SIs for GNN interpretability. Bui et al. (2024) propose an optimal partition of the graph instance as an explanation. This partition is found by approximating second-order STII via Monte Carlo on connected components of the graph, without being structure-aware of GNNs. Note that in our benchmark datasets, there are no isolated components in the graphs, and therefore this does not yield any reduction in complexity. In contrast, GraphSHAP-IQ is able to compute exact STII (and Myerson-STII) for any order.

**GraphSVX (Duval & Malliaros, 2021).** GraphSVX proposes to compute Shapley values without interactions on GNNs. This method does not use any GNN-specific structural assumption for graph classification, since GraphSVX considers SVs (not interactions). For graph classification, GraphSVX is an application of KernelSHAP to GNNs. However, for node classification, a result for SVs based on the dummy axiom was established, which also follows from Lemma 3.5. Note that this result for SVs is not applicable for graph prediction, since the dummy axiom does not hold on the graph level, cf. Section 5.5., global/graph classification in (Duval & Malliaros, 2021). For our main result (Theorem 3.6), considering the purified interactions (MIs) instead of SVs is crucial.

**SubgraphX (Yuan et al., 2021).** SubgraphX identifies isolated subgraphs as explanation. This is done by proposing the SV as a heuristical scoring function for subgraph exploration. Given a subgraph candidate, a reduced game is defined, where the whole subgraph represents a single player. Based on the computed SV, the quality of the subgraph is determined. In contrast, GraphSHAP-IQ does not require to group nodes and does not identify isolated components using a scoring function. Instead, GraphSHAP-IQ computes exact SVs on node level, and SIs on all possible subgraphs up to order $k$, which yields an additive decomposition using all these components, which is faithful to the graph game.

**GStarX (Zhang et al., 2022).** GStarX identifies isolated subgraphs as an explanation. It is related to SubgraphX (Yuan et al., 2021) and proposes an alternative scoring function to the SVs. In contrast, GraphSHAP-IQ does not identify isolated components using a scoring function. Instead, GraphSHAP-IQ decomposes the GNN into all possible components (subgraphs) up to order $k$. Hence, GraphSHAP-IQ explains the GNN prediction across all maskings faithfully according to the SIs.

**SAME (Ye et al., 2023).** SAME proposes a hierarchical MCTS algorithm as a heuristic way to find explanations as subgraphs. SAME considers $k$-hop neighborhoods for computation of the SV, which however is not accurate for graph prediction, similar to GraphSVX (Duval & Malliaros, 2021).

## C.2 LINKING GRAPHSHAP-IQ AND L-SHAPLEY

In this section, we present a link of GraphSHAP-IQ to L-Shapley (Chen et al., 2019) and prove a novel result, which states that L-Shapley with sufficiently large parameter computes exact SVs on games that admit the invariance property (Theorem 3.3), e.g. the GNN-induced node game. L-Shapley is a model-agnostic method to approximate SVs that utilizes the underlying graph structure of features. L-Shapley proposes to compute the marginal contributions $\Delta_i(T \setminus i)$ of subsets $T$ that contain $i \in N$ in its $\lambda$-hop neighborhood

$$\hat{\phi}_\lambda^{\text{L-SV}}(i) := \frac{1}{|\mathcal{N}_i^{(\lambda)}|} \sum_{T \subseteq \mathcal{N}_i^{(\lambda)} : i \in T} \binom{|\mathcal{N}_i^{(\lambda)}| - 1}{t - 1}^{-1} \Delta_i(T \setminus i). \tag{5}$$

The weights thereby match the weights of the SV restricted to its $k$-hop neighborhood. While L-Shapley is restricted to the SV, it strongly differs conceptually from GraphSHAP-IQ in multiple ways. First, GraphSHAP-IQ allows to compute interactions up to order $\lambda$ in the $\ell$-hop neighborhood, whereas L-Shapley (implicitly) computes interactions up to order $\lambda$ in the $\lambda$-hop neighborhood. Second, L-Shapley considers only a single neighborhood of the player $i \in N$, whereas GraphSHAP-IQ includes all interactions that contain player $i$ in any $\ell$-hop neighborhoods. Consequently, GraphSHAP-IQ covers all interactions up to order $\lambda$ of the GNN-induced graph game, whereas L-Shapley only covers interactions up to order $\lambda$ that are fully contained in the $\lambda$-hop neighborhood of the node game of player $i$. Third, L-Shapley is a model-agnostic method, which is based on a heuristical view of the game structure, whereas GraphSHAP-IQ exploits the GNN structure and additivity of all node games summarized in the graph game. Consequently, GraphSHAP-IQ computes exact SV with $\lambda = \ell$, whereas L-Shapley computes exact SVs with $\lambda = n$, i.e. only with all $2^n$ model calls. Lastly, GraphSHAP-IQ maintains efficiency and computes the MIs, which allow to construct any-order SIs yielding an accuracy-interpretability trade-off based on practitioner's needs.

**Theorem C.1.** *For a graph $g$ and a single GNN-induced node game $\nu_i$ for $i \in N$ associated with the node embedding $\mathbf{h}_{v_i}^{(\ell)}$, GraphSHAP-IQ applied on the node game with $\lambda = |\mathcal{N}_i^{(\ell)}|$ and explanation*

*order $k = 1$ returns the L-Shapley computation with $\lambda = \ell$. That is, in this specific case the L-Shapley computation is equal to the GraphSHAP-IQ computation and yields the exact SVs.*

*Proof.* Let $\nu_i$ be a single node game of node $v_i \in V$. If $\lambda = |\mathcal{N}_i^{(\ell)}|$, then by Corollary D.1 GraphSHAP-IQ computes exact MIs contained in $\mathcal{N}_i^{(\ell)}$, which by Lemma 3.5 are all non-trivial interactions of the node game up to order $\lambda = \ell$. Consequently, GraphSHAP-IQ computes exact SVs of the node game. On the other hand, we will show that the L-Shapley computation in Equation (5) also yields these exact SVs in this case. We first observe that due to the node game invariance in Theorem 3.3, we have for

$$T_i^+ := T \cap \mathcal{N}_i^{(\ell)} \subseteq \mathcal{N}_i^{(\ell)} \text{ and } T_i^- := T \cap (N \setminus \mathcal{N}_i^{(\ell)}) \subseteq N \setminus \mathcal{N}_i^{(\ell)},$$

with $T = T_i^+ \sqcup T_i^-$ that

$$\Delta_i(T \setminus i) = \nu(T \setminus i) - \nu(T) = \nu(T_i^+ \setminus i) - \nu(T_i^+) = \Delta_i(T_i^+ \setminus i).$$

Before we compute the SV we need to state the following identity.

**Lemma C.2** (Lemma 1 in Chen et al. (2019)). *For any integer $n$ and pair of non-negative integers $s \geq t$, we have*

$$\sum_{j=0}^{n} \frac{1}{\binom{n+s}{j+t}} \binom{n}{j} = \frac{s+1+n}{(s+1)\binom{s}{t}}.$$

*Proof.* The proof is given in the proof in Lemma 1 in Appendix B by Chen et al. (2019). $\square$

Hence, we can compute the SV as

$$\phi^{\text{SV}}(i) = \sum_{T \subseteq N \setminus i} \frac{1}{n \cdot \binom{n-1}{t}} \Delta_i(T)$$

$$= \sum_{T \subseteq N : i \in T} \frac{1}{n \cdot \binom{n-1}{t-1}} \Delta_i(T \setminus i)$$

$$= \sum_{T \subseteq N : i \in T} \frac{1}{n \cdot \binom{n-1}{t-1}} \Delta_i(T_i^+ \setminus i)$$

$$= \sum_{T_i^+ \subseteq \mathcal{N}_i^{(\ell)} : i \in T_i^+} \Delta_i(T_i^+ \setminus i) \sum_{T_i^- \subseteq N \setminus \mathcal{N}_i^{(\ell)}} \frac{1}{n \cdot \binom{n-1}{t_i^+ + t_i^- - 1}}$$

$$= \sum_{T_i^+ \subseteq \mathcal{N}_i^{(\ell)} : i \in T_i^+} \Delta_i(T_i^+ \setminus i) \sum_{t_i^- = 0}^{n - |\mathcal{N}_i^{(\ell)}|} \binom{n - |\mathcal{N}_i^{(\ell)}|}{t_i^-} \frac{1}{n \cdot \binom{n - |\mathcal{N}_i^{(\ell)}| + (|\mathcal{N}_i^{(\ell)}| - 1)}{t_i^- + (t_i^+ - 1)}}$$

$$= \sum_{T_i^+ \subseteq \mathcal{N}_i^{(\ell)} : i \in T_i^+} \Delta_i(T_i^+ \setminus i) \frac{|\mathcal{N}_i^{(\ell)}| - 1 + 1 + n - |\mathcal{N}_i^{(\ell)}|}{n \cdot (|\mathcal{N}_i^{(\ell)}| - 1 + 1)\binom{|\mathcal{N}_i^{(\ell)}| - 1}{t_i^+ - 1}}$$

$$= \sum_{T_i^+ \subseteq \mathcal{N}_i^{(\ell)} : i \in T_i^+} \Delta_i(T_i^+ \setminus i) \frac{1}{|\mathcal{N}_i^{(\ell)}| \cdot \binom{|\mathcal{N}_i^{(\ell)}| - 1}{t_i^+ - 1}},$$

where we have used Lemma C.2 in the second last equation. This result of the SVs clearly coincides with Equation (5), which concludes the proof. $\square$

Theorem C.1 shows that L-Shapley computes exact SVs, if the game corresponds to a node game in a GNN. However, L-Shapley does not compute exact SVs for the graph game, as interactions involved in the computation of the SV may also appear in other neighborhoods. In fact, L-Shapley applied on the graph game only converges for $k = n$, which corresponds to $2^n$ model evaluations. In our experiments, we also show empirically that L-Shapley performs poorly on GNN-induced graph games.

# D ADDITIONAL ALGORITHMIC DETAILS FOR GRAPHSHAP-IQ

In this section, we provide further details of the GraphSHAP-IQ algorithm, including the full algorithm that is capapable of approximation.

## D.1 EXACT COMPUTATION

Algorithm 1 outlines the exact computation for GraphSHAP-IQ and we provide further details and pseudocode in this section. Algorithm 2 describes the computation of the MIs from line 3 in Algorithm 1. Given the MIs, GraphSHAP-IQ outputs the converted SIs, according to the conversion formulas discussed in Appendix E.3. Algorithm 3 describes the pseudocode of the conversion method called in line 4 in Algorithm 1. Here, the method GETCONVERSIONWEIGHT outputs the distribution weight for each specific index given an MI $\tilde{S}$, a SI $S$ and the index, according to the conversion formulas discussed in Appendix E.3.

---

**Algorithm 2** Möbius Interaction (MI)

**Require:** Game values $\nu_g$ and MI of interest $S \subseteq N$.
1: $m_g[S] \leftarrow 0$
2: **for** $T \in \mathcal{P}(S)$ **do**
3: $\quad m_g[S] \leftarrow m_g[S] + (-1)^{|S|-|T|} \nu_g[T]$
4: **end for**
5: **return** MI $m_g$.

---

**Algorithm 3** MItoSI

**Require:** (Approximated) MIs $\hat{m}_g$, SI order $k$ and SI index
1: **for** $\tilde{S} \in \text{INDEX}(\hat{m}_g)$ **do**
2: $\quad$ **for** $S \in \mathcal{P}(\tilde{S})$ with $|S| \leq k$ **do**
3: $\quad\quad \Phi_k[S] \leftarrow 0$, if not initialized yet.
4: $\quad\quad w_S \leftarrow \text{GETCONVERSIONWEIGHT}(\tilde{S}, S, \text{index})$
5: $\quad\quad \Phi_k[S] \leftarrow \Phi_k[S] + w_S \cdot \hat{m}_g[\tilde{S}]$
6: $\quad$ **end for**
7: **end for**
8: **return** SIs $\Phi_k$.

---

## D.2 APPROXIMATION WITH GRAPHSHAP-IQ

For graphs with high connectivity and GNNs with many convolutional layers, computing exact SIs via Theorem 3.7 may still be infeasible. We thus propose GraphSHAP-IQ, a flexible approach to compute either exact SIs or an approximation. Outlined in Algorithm 4, GraphSHAP-IQ depends on a single parameter $\lambda = 1, \ldots, n$ that controls the size of computed MIs. Given $\lambda$, GraphSHAP-IQ deploys a deterministic approximation method that computes exhaustively all non-trivial MIs of the GNN-induced graph game $\nu_g$ up to order $\lambda$. In Appendix C.2, we discuss a link between SVs and L-Shapley (Chen et al., 2019).

**Exact SIs:** If $\lambda \geq n_{\max}^{(\ell)}$, i.e. $\lambda$ is larger or equal to the largest neighborhood size, then the set of *remaining neighborhoods* $\mathcal{R}_\lambda$ (line 2) is empty. GraphSHAP-IQ then computes all non-trivial MIs and returns exact SIs. In fact, according to Proposition 3.6, $\mathcal{I}_\lambda$ (line 1) contains all non-trivial MIs and GraphSHAP-IQ evaluates the graph game $\nu_g$ on all of these subsets (line 3). The MIs $\hat{m}_g(S)$ are then computed for all $S \in \mathcal{I}_\lambda = \mathcal{I}$ (line 4), and converted to SIs (line 13), cf. Appendix E.3.

**Approximation:** If $\lambda < n_{\max}^{(\ell)}$, then $\mathcal{I}_\lambda$ (line 1) does not contain all non-trivial MIs and the set of remaining neighborhoods $\mathcal{R}_\lambda$ (line 2) is not empty. Consequently, the recovery property of the MIs, cf. Equation (1), does not hold for $T \in \mathcal{R}_\lambda$, and in particular not for the model prediction $\nu_g(N) = f(\mathbf{X})$. Thus, for neighborhoods $T \in \mathcal{R}_\lambda$ GraphSHAP-IQ computes the current recovery value using the MIs (line 7) and assigns the gap to $\hat{m}_g(T)$ (line 8). Note that a previous sorting (line 5) is required to ensure that previously assigned neighborhood MIs are included. Ultimately, the largest neighborhood $S^* \in \mathcal{R}_\lambda$ (line 10) receives the gap $\tau$ (line 11) of the prediction

$\nu_g(N) = f(\mathbf{X})$ and the sum of all MIs to maintain efficiency (line 12). The rationale of maintaining the recovery property is that while GraphSHAP-IQ introduces a bias on the approximated MIs, it equally distributes missing interaction mass onto lower-order SIs. GraphSHAP-IQ further satisfies the following corollary.

**Corollary D.1.** *GraphSHAP-IQ computes exact SIs, if $\lambda \geq n_{\max}^{(\ell)} - 1$.*

*Proof.* By Proposition 3.6 it is clear that GraphSHAP-IQ computes exact MIs and consequently exact SIs for $\lambda \geq \max_{i \in N}(|\mathcal{N}_i^{(\ell)}|)$, since GraphSHAP-IQ computes all MIs in $\mathcal{I}$ up to order $\lambda$. For $\lambda \geq \max_{i \in N}(|\mathcal{N}_i^{(\ell)}|) - 1$, GraphSHAP-IQ computes all MIs up to order $\max_{i \in N}(|\mathcal{N}_i^{(\ell)}|) - 1$ However, for all neighborhoods of size $\max_{i \in N}(|\mathcal{N}_i^{(\ell)}|)$, there is only a single MI remaining, which is then collected in $\mathcal{R}_\lambda$. W.l.o.g let $R$ be the set to the corresponding missing MI. Due to the recovery property of the MIs, this last missing interaction is exactly the gap computed in line 7, since

$$m_g(R) = \sum_{T \subseteq R} (-1)^{r-t} \nu(T) = \nu(R) + \sum_{T \subset R} (-1)^{r-t} \nu(T).$$

Since this holds for all missing interactions in $\mathcal{R}_\lambda$ and further $\mathcal{I} = \mathcal{I}_\lambda \cup \mathcal{R}_\lambda$, it concludes the proof as also the efficiency gap (line 12) is zero due to

$$\sum_{T \in \mathcal{I}_\lambda \cup \mathcal{R}_\lambda} m_g(T) = \sum_{T \in \mathcal{I}} m_g(T) = \sum_{T \subseteq N} m_g(T) = \nu_g(N) = f_g(\mathbf{X}).$$

$\square$

---

**Algorithm 4** GraphSHAP-IQ

---

**Require:** Graph $g = (V, E, \mathbf{X})$, $\ell$-Layer GNN $f_g$, MI order $\lambda$, SI order $k$.
1: $\mathcal{I}_\lambda \leftarrow [T]_{T \in \mathcal{I}, |T| \leq \lambda}$
2: $\mathcal{R}_\lambda \leftarrow [\mathcal{N}_i^{(\ell)}]_{\mathcal{N}_i^{(\ell)} \notin \mathcal{I}_\lambda}$
3: $\nu_g \leftarrow [f_g(\mathbf{X}^{(T)})]_{T \in \mathcal{I}_\lambda \cup \mathcal{R}_\lambda}$
4: $\hat{m}_g \leftarrow [\texttt{MI}(\nu_g, S)]_{S \in \mathcal{I}_\lambda}$
5: $\mathcal{R}_\lambda \leftarrow \texttt{SORT}(\mathcal{R}_\lambda)$
6: **for** $S \in \mathcal{R}_\lambda$ **do**
7: $\quad \hat{\nu}_g[S] \leftarrow \sum_{T \subset S} \hat{m}_g[T]$
8: $\quad \hat{m}_g[S] \leftarrow \nu_g[S] - \hat{\nu}_g[S]$
9: **end for**
10: $S^* \leftarrow \texttt{SELECTMAX}(R_\lambda)$
11: $\tau \leftarrow f_g(\mathbf{X}) - \sum_T \hat{m}_g[T]$
12: $\hat{m}_g[S^*] \leftarrow \hat{m}_g[S^*] - \tau$
13: $\hat{\Phi}_k \leftarrow \texttt{MItoSI}(\hat{m}_g, k)$
14: **return** MIs $\hat{m}_g$, SIs $\hat{\Phi}_k$

---

### D.3 INTERACTION-INFORMED BASELINE METHODS

For a given graph instance and a GNN, Theorem 3.7 states that MIs are sparse, i.e. all MIs of subsets that are not in $\mathcal{I}$ are zero. Unfortunately, model-agnostic baseline methods, such as Permutation Sampling (Tsai et al., 2023), SHAP-IQ (Fumagalli et al., 2023), SVARM-IQ (Kolpaczki et al., 2024b), KernelSHAP-IQ (Fumagalli et al., 2024) or Inconsistent KernelSHAP-IQ (Pelegrina et al., 2023; Fumagalli et al., 2024), use Monte Carlo sampling on the game evaluations, i.e. the masked predictions from the graph game $\nu_g$. Therefore, results on MIs cannot be directly applied to improve the baseline methods. However, all variants of SIs can be represented by MIs, and it has been shown that the SI of a subset $S \subseteq N$ is given as the weighted sum of MIs exclusively of subsets $T \subseteq N$ that contain $S$, i.e. $S \subseteq T$ (Bordt & von Luxburg, 2023). Due to the structure of non-trivial MIs $\mathcal{I}$ from Theorem 3.6, the following holds: For two sets $S, T \subseteq N$ with $S \subseteq T$, if $T \in \mathcal{I}$, then also $S \in \mathcal{I}$. Conversely, if $S \notin \mathcal{I}$, then also $T \notin S$. In other words, for the SI of a subset $S \subseteq N$ with $S \notin \mathcal{I}$, its MI is zero, as well as all MIs of subsets $T$ that contain $S$. Hence, the SI for a subset $S \notin \mathcal{I}$ is

zero as well, since it can be computed as a weighted sum exclusively of MIs of subsets that contain $S$. In conclusion, SIs of subsets that are not contained in $\mathcal{I}$ can be discarded from approximation and set to zero. For baseline methods that estimate each SI separately using Monte Carlo, such as Permutation Sampling, SHAP-IQ and SVARM-IQ, we exclude these SIs from approximation and set their values to zero. For regression-based methods, such as KernelSHAP-IQ and Inconsistent KernelSHAP-IQ, these interactions are excluded from the regression problem, and manually set to zero.

Notably, since all individual nodes are contained in $\mathcal{I}$, i.e. all nodes affect the graph prediction, the approximation of the SV is unaffected by this adjustment. Consequently, we do not adjust L-Shapley (Chen et al., 2019), as it is only applicable for SVs.

# E  Further Background on Shapley Interactions

In this section we provide further background on SIs. We first provide more related work in Appendix E.1. In Appendix E.2, we introduce SIs that satisfy the efficiency axiom. In Appendix E.3, we discuss conversion formulas of MIs to various SIs.

## E.1  Related Work on Shapley Interactions

The SV (Shapley, 1953) is a concept from game theory (Grabisch, 2016) and applied in XAI for local (Lundberg & Lee, 2017; Covert et al., 2021; Chen et al., 2023) and global (Casalicchio et al., 2019; Covert et al., 2020) model interpretability, or data valuation (Ghorbani & Zou, 2019). The Myerson value (Myerson, 1977) is a variant of the SV for games restricted on components of graphs. Extensions of the SV to interactions were proposed in game theory (Grabisch & Roubens, 1999; Marichal & Roubens, 1999) and ML (Lundberg et al., 2020; Sundararajan et al., 2020; Tsai et al., 2023; Bordt & von Luxburg, 2023), where ML-based extensions preserve efficiency, e.g. the sum of interactions equals the model's prediction. The MIs (Rota, 1964; Harsanyi, 1963; Fujimoto et al., 2006) define a fundamental concept in game theory, and the edge case of ML-based SIs. Due to the exponential complexity of SIs, efficient methods for tree-based models for SVs (Lundberg et al., 2020; Yu et al., 2022), any-order SIs (Zern et al., 2023; Muschalik et al., 2024b) and MIs (Hiabu et al., 2023) were proposed. In model-agnostic settings, approximation methods have been proposed (Tsai et al., 2023; Fumagalli et al., 2023; Kolpaczki et al., 2024b; Fumagalli et al., 2024) as extensions of SV-based methods (Castro et al., 2009; Covert & Lee, 2021; Kolpaczki et al., 2024b; Lundberg & Lee, 2017; Pelegrina et al., 2023), and by exploiting graph-structured inputs (Chen et al., 2019). Recently, MIs were computed in synthetic settings (Kang et al., 2024).

## E.2  Efficient Shapley Interaction Indices

In the following, we describe extensions of the SV to SIs, which yield efficient interaction indices. That is, given an explanation order $k$ the SIs $\Phi_k$ are defined on all interactions up to order $k$ and yield an additive decomposition of the model's prediction, i.e

$$\nu(N) = \sum_{S \subseteq N, |S| \leq k} \Phi_k(S).$$

**$k$-Shapley Values ($k$-SII)**  The $k$-SIIs values of order 1 are defined as the SVs $\phi^{\text{SV}}$. Furthermore, the pairwise $k$-SIIs are given as

$$\Phi_2(ij) := \phi^{\text{SII}}(ij) \qquad \text{and} \qquad \Phi_2(i) := \phi^{\text{SV}}(i) - \frac{1}{2}\sum_{j \in N \setminus i} \phi^{\text{SII}}(ij).$$

The general case is described by Bordt & von Luxburg (2023) and involves the Bernoulli numbers. For any explanation order $k = 1, \ldots, n$, the $k$-SIIs are recursively defined as

$$\Phi_k(S) := \begin{cases} \phi^{\text{SII}}(S) & \text{if } |S| = k \\ \Phi_{k-1}(S) + B_{k-|S|} \sum_{\tilde{S} \subseteq N \setminus S}^{|S| + \tilde{S} = k} \phi^{\text{SII}}(S \cup \tilde{S}) & \text{if } |S| < k \end{cases}$$

with $1 \leq |S| \leq k \leq n$, $\Phi_0 \equiv 0$ and Bernoulli numbers $B_n$ (Bordt & von Luxburg, 2023). For $k = 1$, $k$-SII is the SV, and for $k = n$ the MI (Bordt & von Luxburg, 2023). An explicit definition of $k$-SII is given in Appendix A.1 in Bordt & von Luxburg (2023) as

$$\Phi_k(S) = \sum_{r=0}^{k-|S|} \sum_{R \subseteq N, |R| = r} B_k \phi^{\text{SII}}(S \cup R).$$

Besides SII and $k$-SII, there exist other interaction indices that extend the SV to higher orders and yield the MIs for $k = n$. These essentially differ by their distribution of MI to lower-order interactions. Similar to $k$-SII, these interaction indices can be described by the MIs. Hence, they can be computed once the MIs are known.

**Shapley Taylor Interaction Index (STII)**   The STII (Sundararajan et al., 2020) is an interaction index that gives stronger emphasis to the highest order of computed interactions, which capture all higher-order MIs. It is defined as

$$\Phi_k^{\text{STII}}(S) := a(S) \text{ for } |S| < k \qquad \text{and} \qquad \Phi_k^{\text{STII}}(S) := \frac{k}{n} \sum_{T \subseteq N \setminus S} \frac{1}{\binom{n-1}{|T|}} \Delta_S(T) \text{ for } |S| = k.$$

The STII for lower order, i.e. $|S| < k$, is simply the MI. For the maximum order, i.e. $|S| = k$, the STII is computed as an average over discrete derivatives. Therefore, the STII distributes the higher-order MI with $|S| > k$ solely on the top-order interactions of the STII. It was shown (Sundararajan et al., 2020) that the STII is the only interaction index preserving the Shapley axioms that additionally satisfies the *interaction distribution axiom* besides the classical axioms.

**Faithful Shapley Interaction Index (FSII)**   The FSII (Tsai et al., 2023) is an interaction index that yields stronger emphasis on the faithfulness of the interaction index. It is defined as the best approximation of interactions up to order $k$ in terms of a weighted least square objective

$$\Phi_k^{\text{FSII}} := \underset{\Phi_k \subseteq \mathbb{R}^{n_k}}{\arg\min} \sum_{T \subseteq N,\ 0 < |T| < n} \mu(t) \left( \nu(T) - \sum_{S \subseteq T, |S| \le k} \Phi_k(S) \right)^2,$$

$$\text{such that } \nu(N) = \sum_{S \subseteq N,\ |S| \le k} \Phi_k(S) \text{ and } \nu(\emptyset) = \Phi_k(\emptyset),$$

where $\mu(t) \propto \frac{n-1}{\binom{n}{t} \cdot t \cdot (n-t)}$ and $n_k := |\{S \subseteq N \mid |S| \le k\}|$ is the number of interactions up to size $k$. It was shown that the FSII is the only interaction index preserving the Shapley axioms that additionally satisfies the *faithfulness property*, i.e. is represented as the solution to a single weighted least square problem (Tsai et al., 2023).

### E.3   CONVERSION FORMULAS FOR MÖBIUS INTERACTIONS TO SHAPLEY INTERACTIONS

The MIs are a basis of the vector space of cooperative games and thus all SIs can be expressed in terms of the MIs. The SV as well as the SII can be directly computed by the conversion formulas

$$\phi^{\text{SV}}(i) = \sum_{\tilde{S} \subseteq N : i \in \tilde{S}} \frac{m(\tilde{S})}{|\tilde{S}|} \qquad \text{and} \qquad \phi^{\text{SII}}(S) = \sum_{\tilde{S} \subseteq N : \tilde{S} \supseteq S} \frac{m(\tilde{S})}{|\tilde{S}| - |S| + 1}.$$

The conversion formula for the MI to $k$-SII is given in Theorem 6 in Bordt & von Luxburg (2023). The conversion for STII is given in the proof of Theorem 8 in Appendix H (Bordt & von Luxburg, 2023). The conversion formula from to the FSII is given in Theorem 19 in Tsai et al. (2023).

## F    Experimental Setup and Reproducibility

All the experiments have been conducted using the PyTorch Geometric library (Fey & Lenssen, 2019), running on a computing machine equipped with Intel(R) Xeon(R) CPU, one Nvidia RTX A5000 and 60GB of RAM. Overall, the total compute time of this project, including preliminary experiments, training of GNNs, computation of baselines and SIs, required no more than 100 hours, which could be reduced by simple parallelization.

### F.1    Datasets

All considered methods were empirically validated on eight common real-world chemical datasets for graph classification and one real-world water distribution network for graph regression. We avoid synthetic datasets commonly used in the graph XAI community due to their limitations (Amara et al., 2022). The licenses for the datasets are summarized in Table 3.

Table 3: Dataset License Overview

| Dataset | Source | License |
|---|---|---|
| BNZ | (Sánchez-Lengeling et al., 2020) | CC0 1.0 Universal |
| FLC | (Sánchez-Lengeling et al., 2020) | CC0 1.0 Universal |
| MTG | (Kazius et al., 2005) | CC0 1.0 Universal |
| ALC | (Sánchez-Lengeling et al., 2020) | CC0 1.0 Universal |
| PRT | (Borgwardt et al., 2005; Morris et al., 2020) | "Free to use", cf. Section 2 of Morris et al. (2020) |
| ENZ | (Borgwardt et al., 2005; Morris et al., 2020) | "Free to use", cf. Section 2 of Morris et al. (2020) |
| CX2 | (Sutherland et al., 2003; Morris et al., 2020) | "Free to use", cf. Section 2 of Morris et al. (2020) |
| BZR | (Sutherland et al., 2003; Morris et al., 2020) | "Free to use", cf. Section 2 of Morris et al. (2020) |
| WAQ | (Vrachimis et al., 2018) | EUROPEAN UNION PUBLIC LICENCE v. 1.2 EUPL |

#### F.1.1    Chemical Datasets

This data reflects biological and chemical problems, where particular substructures can predict the properties of molecules. The **Benzene (BNZ)** (Sánchez-Lengeling et al., 2020) dataset consists of 12 000 moleculars, with each graph labeled as containing or not a benzene ring. In this context, the underlying explanations for the predictions are the specific atoms (nodes) that make up the benzene rings. The **Fluoride Carbonyl (FLC)** (Sánchez-Lengeling et al., 2020) dataset comprises 8 671 molecular graphs, each labeled positive or negative. A positive sample denotes a molecule with a fluoride ion (F-) and a carbonyl functional group (C = O). The underlying explanations for the labels are the specific combinations of fluoride atoms and carbonyl functional groups present within a given molecule. Similarly, the **Alkane Carbonyl (ALC)** (Sánchez-Lengeling et al., 2020) dataset consists of 1 125 molecular graphs, labeled positive if it features an unbranched alkane chain and a functional carbonyl group (C = O), which serves as an underlying explanation. The **Mutagenicity (MTG)** (Kazius et al., 2005) dataset comprises 1 768 graph molecules, categorized into two classes based on their mutagenic properties, specifically their effect on the Gram-negative bacterium S. Typhimurium. Following Agarwal et al. (2023), the original dataset of 4 337 graphs was reduced to 1 768, retaining only those molecules whose labels directly correlate with the presence or absence of specific toxicophores (motifs in the graphs), including NH2, NO2, aliphatic halide, nitroso, and azo-type groups, as defined by (Kazius et al., 2005). The latter four datasets have been retrieved from the implementation of (Agarwal et al., 2023). **PROTEINS (PRT)** (Borgwardt et al., 2005) and **ENZYMES (ENZ)** (Borgwardt et al., 2005) involve graphs whose nodes represent secondary structure elements and edges indicate neighborhood in the amino-acid sequence, entailing 1 113 and 600 graphs respectively. **BZR (BZR)** and **COX2 (CX2)** (Sutherland et al., 2003) are correlation networks of biological activities of compounds with their structural attributes, of 405 and 467 graphs. These latter datasets are fetched from the benchmark collection of (Morris et al., 2020). They all have binary labels except for ENZ, which has 6 classes. Finally, we did not exploit the additional continuous node features available for ENZ.

#### F.1.2    Water Distribution Network

We apply GraphSHAP-IQ to a dynamical system governed by local partial differential equations (PDEs). Specifically, the spread of chlorine in water distribution systems (WDS), where a GNN pre-

Table 4: Model accuracy for shallow and deep GNNs (1-6 layers).

| Dataset | Dataset Description | | | | Model Accuracy by Layer (%) | | | | | | | | | | | | | | | | | |
| | Graphs | $d_{out}$ | Nodes (avg) | Density (avg) | GCN | | | | | | GAT | | | | | | GIN | | | | | |
| | | | | | 1 | 2 | 3 | 4 | 5 | 6 | 1 | 2 | 3 | 4 | 5 | 6 | 1 | 2 | 3 | 4 | 5 | 6 |
| BNZ (Sánchez-Lengeling et al., 2020) | 12000 | 2 | 20.6 | 22.8 | 84.2 | 88.6 | 90.4 | **90.6** | 90.6 | **90.6** | 83.1 | 85.1 | 85.7 | 84.5 | **87.8** | 86.3 | 84.9 | 90.5 | 90.8 | 92.1 | **92.3** | **92.3** |
| PRT (Borgwardt et al., 2005) | 1113 | 2 | 39.1 | 42.4 | 75.2 | 71.1 | **74.0** | 70.3 | 70.3 | 69.4 | 75.3 | 60.5 | **79.8** | 69.4 | 68.5 | 68.5 | **79.3** | 74.9 | 67.7 | 72.1 | 73.0 | 73.9 |
| ENZ (Borgwardt et al., 2005) | 600 | 6 | 32.6 | 32.0 | 34.2 | 37.5 | 35.8 | **43.3** | 40.0 | 31.7 | 32.5 | 35.0 | 35.8 | 23.3 | 43.3 | **45.0** | 36.7 | 35.0 | **39.2** | 25.0 | 30.0 | 30.0 |

dicts the fraction of nodes chlorinated after some time, framing this task as a graph-level regression. We chose this task because chemicals in WDS spread predominantly with water flow (advection). Therefore, the task should be mainly explained by the flow pattern of the network, which yields an intuitive expectation of the explanations.

We generate a temporal dataset **WaterQuality (WAQ)** using the EPyT-Flow toolbox (Artelt et al., 2024). The dataset consists of 1000 temporal graphs with 30 time steps each. Spatially, the graphs represent the Hanoi WDS (Vrachimis et al., 2018), a popular 32-node benchmark WDS.

## F.2 GNN ARCHITECTURES

Our approach universally applies to any message passing GNN and can be employed on graph datasets without requiring ground truth explanations. Since different GNN architectures exhibit unique learning patterns and performance characteristics, their explanations will also vary. To demonstrate the effectiveness of our approach, we followed the remarks and setting of Amara et al. (2022), by comparing the explanations of three popular GNN models: GCN (Kipf & Welling, 2017), GIN (Xu et al., 2019), and GAT (Velickovic et al., 2018). The overall model architecture is as follows: the graph input is unprocessed, we stack the GNN layers for $l \in \{1, 2, 3\}$, concatenate all the embeddings (You et al., 2020), apply global sum-pooling (Xu et al., 2019), and finally, a dense linear layer returns the output predictions. Furthermore, each GNN layer is followed by a sequence of modules: `batch_normalization`, `dropout`, and `LeakyReLU` activation function (You et al., 2020). We validated the models for different amounts of hidden units $h \in \{8, 16, 32, 64, 128\}$, and we defined the multilayer perceptron of GIN with 32 hidden units, while all the remaining design choices for the GNN layers follow their default settings (e.g., aggregation function, number of attention heads, ...). For a selection of datasets, we also compared performances for deeper GNNs, cf. Table 4. The performances of the deeper models compared to more shallow GNNs are mixed. All architectures lead to quite similar performances. Both shallow (less than 4 GNN layers) and deeper models (more than 3 GNN layers) lead to best performing architectures.

Regarding WAQ, since we now have important edge features (water flows), we cannot use vanilla GCN, GIN, or GAT as before. Instead, we designed a simple GNN that can process edge features, using 3 message-passing layers with mean-aggregation, `ReLU` activations, global average-pooling as the readout layer, and finally, a linear layer provides the regression outputs.

## F.3 TRAINING

The adopted loss is the *(binary) Cross-Entropy* for classification and the $L_1$ for the regression. The chemical datasets have been split in train/validation/test sets with a ratio of $80/10/10$ in a stratified fashion, ensuring that the splits have a balanced amount of classes. The models are randomly initialized and trained for 500 epochs with `early_stopping` with patience of 100 on the validation set and a batch size of 64. The models have been trained with *Adam* optimizer with a starting learning rate of 0.01, using a halving scheduler with the patience of 50 epochs up to $1e^{-5}$, and `weight_decay` factor of $5e^{-4}$.

The test accuracies for the best-validated model configuration reported in Table 1, are comparable to existing benchmarks (Errica et al., 2020; You et al., 2020), confirming that our approach is valid for complex and powerful GNN architectures, as long as Assumption 3.4 holds. Moreover, deeper GNNs do not necessarily yield better performances, cf. Table 4.

# G ADDITIONAL EXPERIMENTAL RESULTS

This section includes additional empirical results and further details on the experimental setup. Appendix G.1 contains additional complexity results for benchmark datasets. Appendix G.3 shows a timeline for the WDN example use case. Appendix G.4 shows the approximation quality of GraphSHAP-IQ compared to a collection of state-of-the-art baseline methods. Appendix G.5 showcases example SIs on molecule structures from different datasets.

## G.1 COMPLEXITY ANALYSIS

In this experiment we provide further results on Section 4.1. We evaluate the complexity on GNNs with varying number of convolutional layers on multiple benchmark datasets, as described in Table 1. The complexity is measured by number of evaluations of the GNN-induced graph game, i.e. number of model calls of the GNN, which is the limiting factor of SIs (Fumagalli et al., 2023; Kolpaczki et al., 2024b; Muschalik et al., 2024b). Due to Theorem 3.7, the complexity of GraphSHAP-IQ does only depend on the message passing ranges and is in particular independent of the message passing technique. We therefore omit the explicit architecture in this experiment and focus only on the number of convolutional layers $\ell$. For every graph in the benchmark datasets, we evaluate the complexity of GraphSHAP-IQ, where the model-agnostic method requires $2^n$, where $n$ is the number of nodes in the graph. If the size of the largest neighborhood exceeds 22, i.e. GraphSHAP-IQ requires at least $2^{23} \approx 8.3 \times 10^6$ model calls, then the complexity is approximated by the upper bound presented in Theorem 3.7. Figure 7, Figure 8, and Figure 9 display the complexity (in $\log 10$) by the size of the graphs for all instances (upper) and the median, Q1, and Q3 per graph size ($n$) for all benchmark datasets. The model-agnostic computation is represented by the dashed line. The results show that the ground-truth computation of SIs is substantially reduced by GraphSHAP-IQ for many instances in the benchmark datasets.

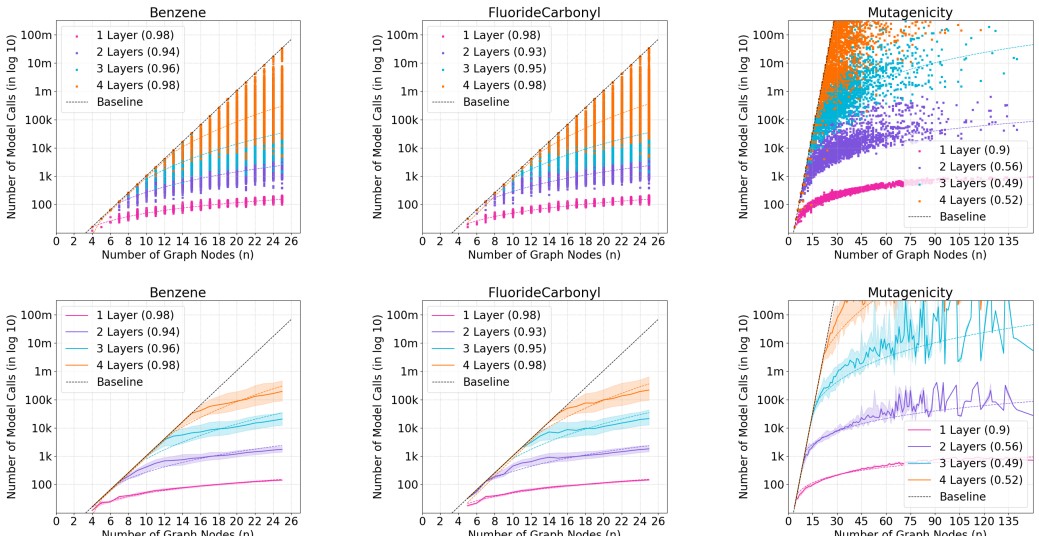

Figure 7: Complexity of GraphSHAP-IQ against model-agnostic baseline (dashed) in model calls (in $\log 10$) (y-axis) by number of nodes (x-axis) for all instances (upper) and median, Q1, Q3 (lower) for *BNZ* (left), *FLC* (middle), and *MTG* (right).

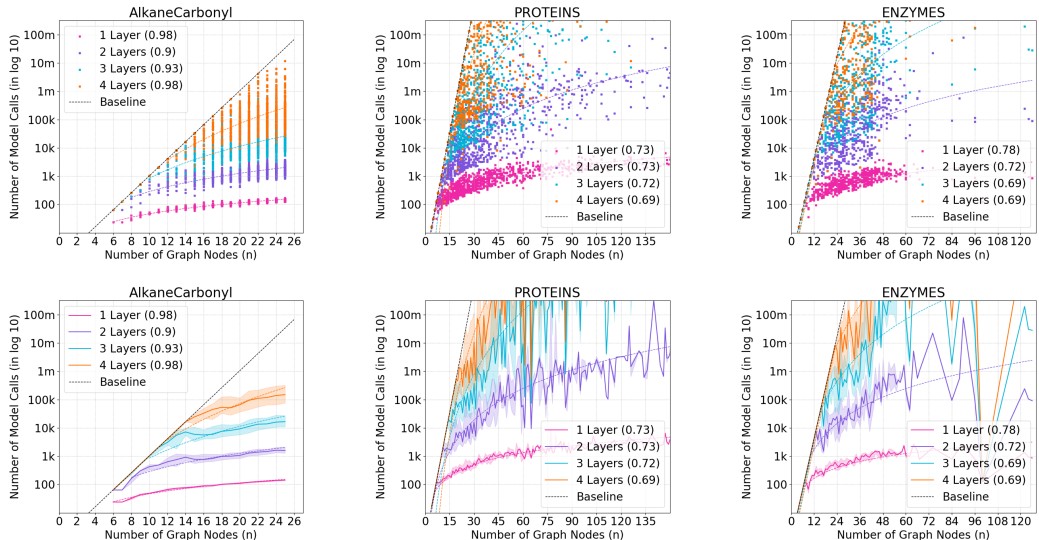

Figure 8: Complexity of GraphSHAP-IQ against model-agnostic baseline (dashed) in model calls (in $\log 10$) (y-axis) by number of nodes (x-axis) for all instances (upper) and median, Q1, Q3 (lower) for *ALC* (left), *PRT* (middle), and *ENZ* (right).

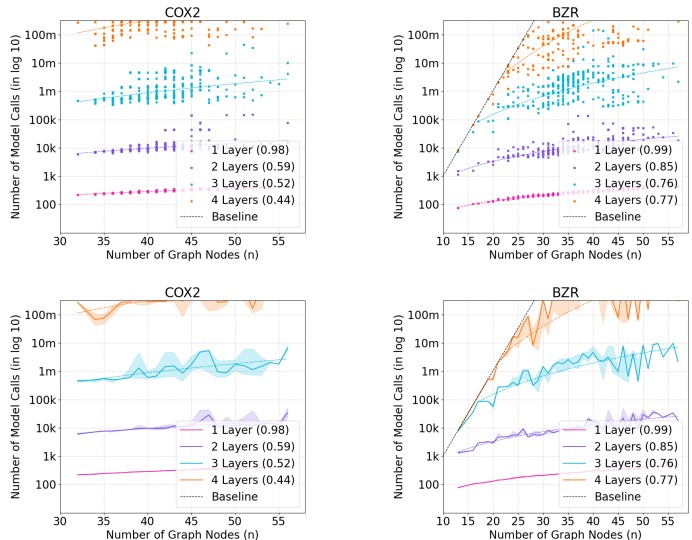

Figure 9: Complexity of GraphSHAP-IQ against model-agnostic baseline (dashed) in model calls (in $\log 10$) (y-axis) by number of nodes (x-axis) for all instances (upper) and median, Q1, Q3 (lower) for *CX2* (left) and *BZR* (right).

## G.2 RUNTIME ANALYSIS

In this section, we conduct a small runtime analysis to confirm that the main driver of computational complexity for GraphSHAP-IQ is indeed the number of model calls. This is standard in the approximation literature of SIs (Tsai et al., 2023) and confirmed for all baseline methods (Fumagalli et al., 2024). We select 100 graph instances from *MTG* with 20 to 40 nodes, where GraphSHAP-IQ requires less than 10000 model calls for exact computation for a 2-Layer GCN. We then compute exact SIs for order 1 (SV), 2 (2-SII), and 3 (3-SII) with GraphSHAP-IQ. For each graph instance, we run all baseline methods using the same budget that GraphSHAP-IQ required for exact computation. Figure 10 shows the number of model calls and runtime (upper row), as well as number of graph nodes and runtime (lower row) for order 1 (left), 2 (middle), and 3 (right). As expected, GraphSHAP-IQ's runtime scales linearly with number of model calls, and is basically unaffected by the size of the graph. Moreover, given the same number of model calls, GraphSHAP-IQ's runtime is similar to efficient baselines (SHAP-IQ and Permutation Sampling). Notably, increasing the explanation order barely affects the runtime of GraphSHAP-IQ, whereas it substantially increases the runtime of competitive baselines (KernelSHAP-IQ, SVARM-IQ, SHAP-IQ).

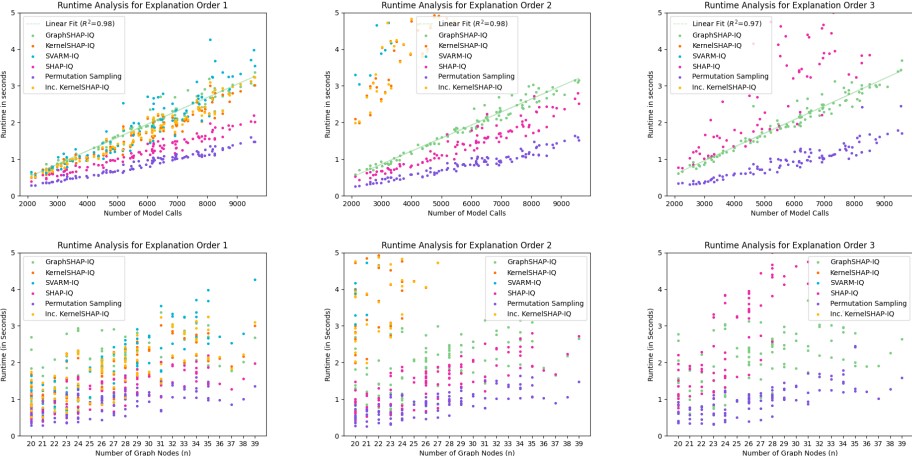

Figure 10: Runtime comparison of GraphSHAP-IQ and baseline methods for SVs (left), 2-SII (middle), and 3-SII (right). All methods were given the number of model calls, which were required for exact computation with GraphSHAP-IQ. The runtime (y-axis) against number of model calls (x-axis) plot (upper row) shows that the runtime of GraphSHAP-IQ scales linearly ($R^2 > 0.97$) with increasing number of model calls. In contrast, the runtime (y-axis) against number of graph nodes (x-axis) plot (lower row) shows that the runtime is independent from the size of the graph. Lastly, increasing the explanation order substantially increases the runtime of most baseline methods, but barely affects GraphSHAP-IQ.

### G.3 Explaining Water Quality in Water Distribution Networks (WDNs)

We investigate the validity of our approach in explaining a critical real-world scenario: adding chlorine to water distribution system (WDS) is a common disinfection practice used to inactivate microorganisms that can cause waterborne diseases. Chlorine is an effective disinfectant due to its ability to oxidize and denature microbial cells. However, it can be toxic to aquatic organisms, particularly at high concentrations, and can impart an unpleasant taste or odor to the water, affecting consumer acceptance. In practice, chlorine is added to a water *reservoir* and flows unevenly through the network. Therefore, it is essential to model these flows and concentrations, which is not trivial since it is a dynamical system governed by local PDEs. We frame this problem as graph regression task, where the goal is to predict the relative chlorine concentration for each node at each time step. In Figure 11, we report the Hanoi WDN (Vrachimis et al., 2018) and simulate a chlorine injection at node reservoir 31 for $t = 0$. We observe how the 2-SII highlight the progressive importance of the nodes — from the reservoir throughout the network.

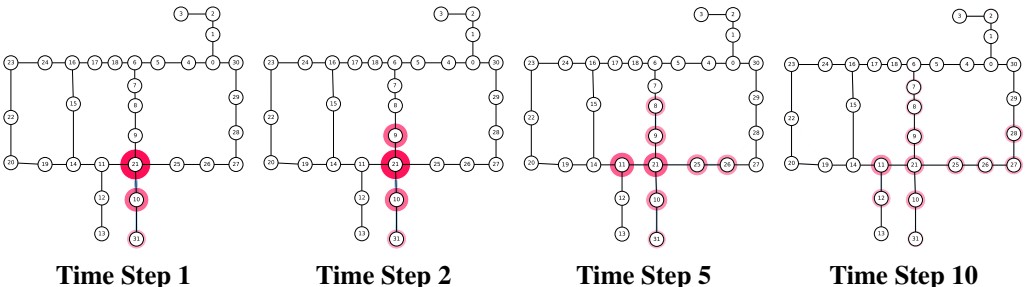

**Time Step 1**    **Time Step 2**    **Time Step 5**    **Time Step 10**

Figure 11: Spread of chlorination through a WDN over time as explained by 2-SII.

### G.4 Approximation Quality of GraphSHAP-IQ

Additionally to the experiments conducted in Section 4.2, this section contains further evaluations on three datasets and a detailed description of the SV baselines (Appendix G.4.2) and the 2-SII baselines (Appendix G.4.3). Figures 12 to 14 show the approximation quality of different GNN architectures for the *MTG*, *PRT*, and *BZR* benchmark datasets. Appendix G.4.1 outlines the approximation experiments in detail.

#### G.4.1 Setup Description

To compare GraphSHAP-IQ's approximation result with existing model-agnostic baselines, we compute SIs for three benchmark datasets (*MTG*, *PRT*, and *BZR*). For each dataset, we randomly select 10 graphs containing $30 \leq n \leq 40$ nodes each. Based on Section 4.1, we further limit the selection to graphs which can be exactly computed via GraphSHAP-IQ with at most $10\,000$ (for *MTG* and *PRT*) and $2^{15} = 32\,768$ (for *BZR*) model evaluations. First, we compute SIs with GraphSHAP-IQ for each MI interaction order $\lambda$ starting with $\lambda = 1$ until $\lambda \geq \max_{i \in N}(|\mathcal{N}_i^{(\ell)}|) - 1$ (c.f. Corollary D.1). Note that for different graphs even for the same datasets the maximum $\lambda$ value may differ. For each $\lambda$, we observe the required model evaluations by GraphSHAP-IQ. Second, we compute SIs with the baseline methods provided the same approximation budget as required by GraphSHAP-IQ for each $\lambda$. Therein, we directly compare how the baselines (which may be run with any arbitrary approximation budget) compare with GraphSHAP-IQ. For each estimation budget (number of model evaluation), we estimate the SIs via two independent iterations for each baseline and average the evaluation metrics (each dot in Figures 12 to 15 are averaged evaluation metrics over two runs). As evaluation metric we choose the mean-squared-error (MSE, lower is better). For each graph, we compute the MSE between the ground truth SIs, as computed by exact GraphSHAP-IQ, and the estimated values.

#### G.4.2 Description of SV Baselines

As SV baselines we apply current state-of-the-art sampling based methods that operate on different representations of the SV and/or aggregation technique. In general, all of the following baseline

methods (with L-Shapley as the only exception) are sampling coalitions of players, evaluate the model on these coalitions, observe the output of the model, and finally aggregate the observed outputs into the SV. All SV baselines are implemented according to the *shapiq* (Fumagalli et al., 2023) open-source software package in Python.

**KernelSHAP** as proposed in (Lundberg & Lee, 2017) is a prominent SV approximation method and is applied akin to its original conception in addition to the sampling tricks discussed in (Covert & Lee, 2021; Fumagalli et al., 2023).

**k-Additive KernelSHAP** (Pelegrina et al., 2023) is an extension to KernelSHAP, which also utilizes higher-order SIs in the computation procedure of the SV. Therein, it shows better approximation qualities than KernelSHAP. For our experiments, we set the higher-order interactions to 2.

**Unbiased KernelSHAP** (Covert & Lee, 2021) extends on KernelSHAP and offers a provably unbiased alternative. It was recently shown that Unbiased KernelSHAP is linked to the $k$-SII SHAP-IQ estimator (Fumagalli et al., 2023).

**Permutation Sampling** for SV (Castro et al., 2009) is a standard estimation method that iterates over random permuations of the player set to determine the coalitions to be used for the model evaluations. Because a permutation always needs to be traversed in its entirety, the number of model calls may be lower than GraphSHAP-IQ's or the rest of the baselines.

**SVARM** (Kolpaczki et al., 2024a) is another sampling-based baseline operating on a stratified representation of the SVs. Akin to KernelSHAP it can be queried on an arbitrary number of coalitions.

**L-Shapley** (Chen et al., 2019) is a deterministic method for computing the Shapley values for structured data. Unlike the rest of the SV approximation methods, L-Shapley cannot be evaluated on an arbitrary set of coalitions or number of model evaluations. L-Shapley functions similarly to GraphSHAP-IQ in that it deterministically evaluates coalitions based on neighborhoods in the graphs. To circumvent this, we let L-Shapley akin to GraphSHAP-IQ. After L-Shapley exceeds GraphSHAP-IQ's number of model evaluation we stop the iteration and use the last estimates (the estimates where L-Shapley always exceeds GraphSHAP-IQ in terms of model evaluations).

### G.4.3 DESCRIPTION OF 2-SII BASELINES

Similar to Appendix G.4.2, we apply current state-of-the-art sampling based methods for 2-SII and in general $k$-SII. All of the following baseline methods are sampling-based. We implement the experiments based on the *shapiq* (Fumagalli et al., 2023) open-source Python software package.

**KernelSHAP-IQ** (Fumagalli et al., 2024) directly extends the SV KernelSHAP approximation paradigm to SII and, hence, to $k$-SII. KernelSHAP-IQ leads to high quality estimates.

**Inconsistent KernelSHAP-IQ** (Fumagalli et al., 2024) is a different version of KernelSHAP-IQ, which is linked to the k-Additive KernelSHAP (Pelegrina et al., 2023). While often leading to better estimations in lower number of model evaluations, Inconsistent KernelSHAP-IQ does not converge to the ground-truth values like KernelSHAP-IQ.

**SHAP-IQ** (Fumagalli et al., 2023) is a sampling-based mean estimator for computing SIIs among other interaction indices like STII. It is theoretically linked to Unbiased KernelSHAP (Fumagalli et al., 2023) and, thus, transfers its SV procedure to SII.

**Permutation Sampling** for SII(Tsai et al., 2023) directly transfers the permutation sampling procedure from its SV counterpart (Castro et al., 2009) to estimate SII. Similar to the SV algorithm, the SII variant requires a full pass through a permutation which may lead to less model evaluations with this baseline.

**SVARM-IQ** (Kolpaczki et al., 2024b) extends SVARM's (Kolpaczki et al., 2024a) stratified representation from the SV to the SII. Often SVARM-IQ outperforms state-of-the-art approximation methods.

### G.5 EXPLANATION GRAPHS FOR MOLECULE STRUCTURES

This section contains further exemplary SI-graphs for molecule structures of the *MTG* and *BNZ* datasets. Figure 16 shows two molecules from the *MTG* dataset, where one of those molecules is

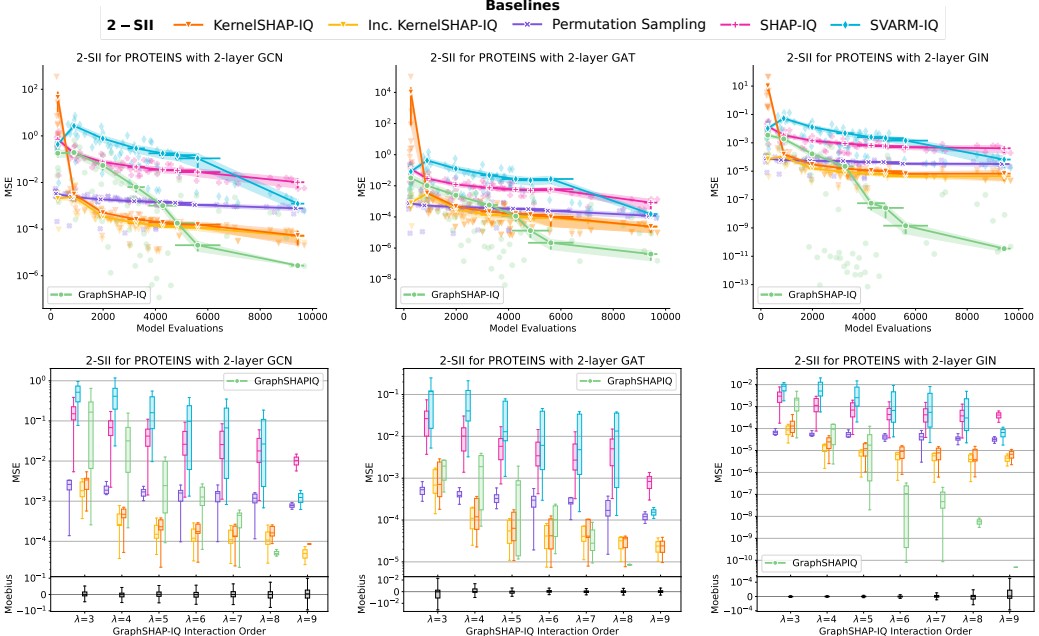

Figure 12: Comparison of GraphSHAP-IQ's approximation quality with model-agnostic baselines on 10 graphs with $30 \leq n \leq 40$ nodes from the *PRT* dataset for a 2-layers GCN (left), GAT (middle) and GIN (right). The top row presents the MSE for each estimation (dots) and averaged over $\lambda$ (line) with the standard error of the mean (confidence band). The bottom row shows the same information including the MIs as box plots for each $\lambda$.

the same as in Figure 1. Further, Figure 17 shows additional SI-Graphs for molecules of the *BNZ* dataset. Lastly, Figure 18 shows how the SI-Graphs differ for different GNN architectures.

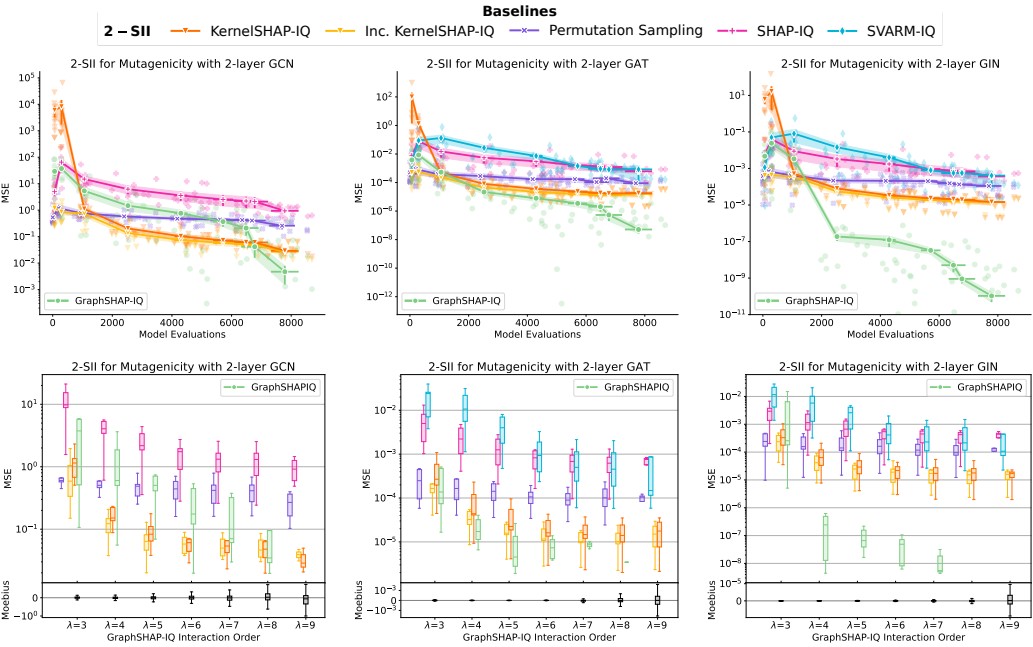

Figure 13: Comparison of GraphSHAP-IQ's approximation quality with model-agnostic baselines on 10 graphs with $30 \leq n \leq 40$ nodes from the *MTG* dataset for a 2-layers GCN (left), GAT (middle) and GIN (right). The top row presents the MSE for each estimation (dots) and averaged over $\lambda$ (line) with the standard error of the mean (confidence band). The bottom row shows the same information including the MIs as box plots for each $\lambda$.

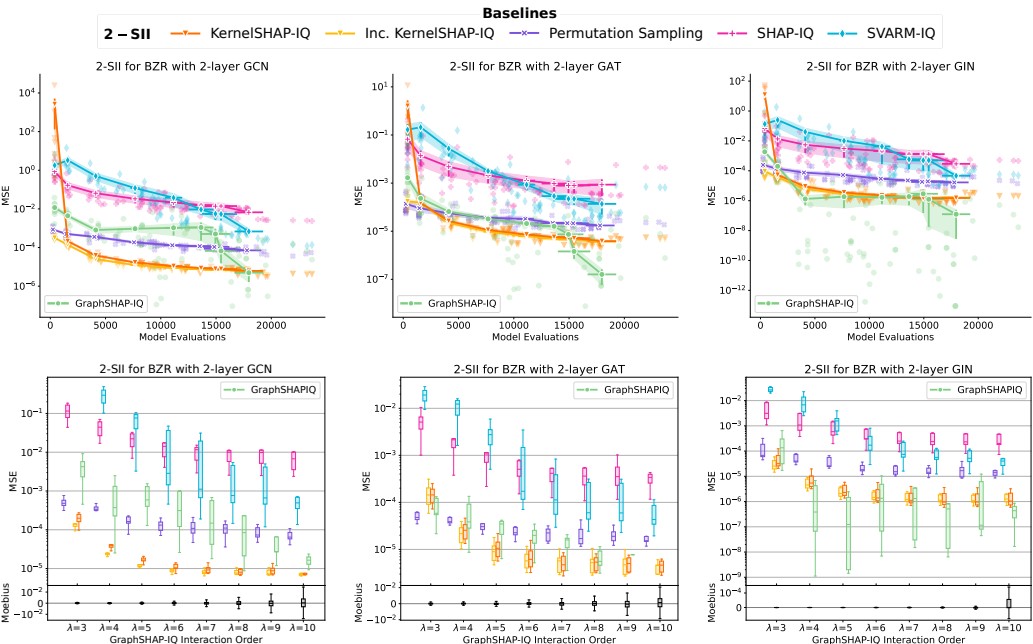

Figure 14: Comparison of GraphSHAP-IQ's approximation quality with model-agnostic baselines on 10 graphs with $30 \leq n \leq 40$ nodes from the *BZR* dataset for a 2-layers GCN (left), GAT (middle) and GIN (right). The top row presents the MSE for each estimation (dots) and averaged over $\lambda$ (line) with the standard error of the mean (confidence band). The bottom row shows the same information including the MIs as box plots for each $\lambda$.

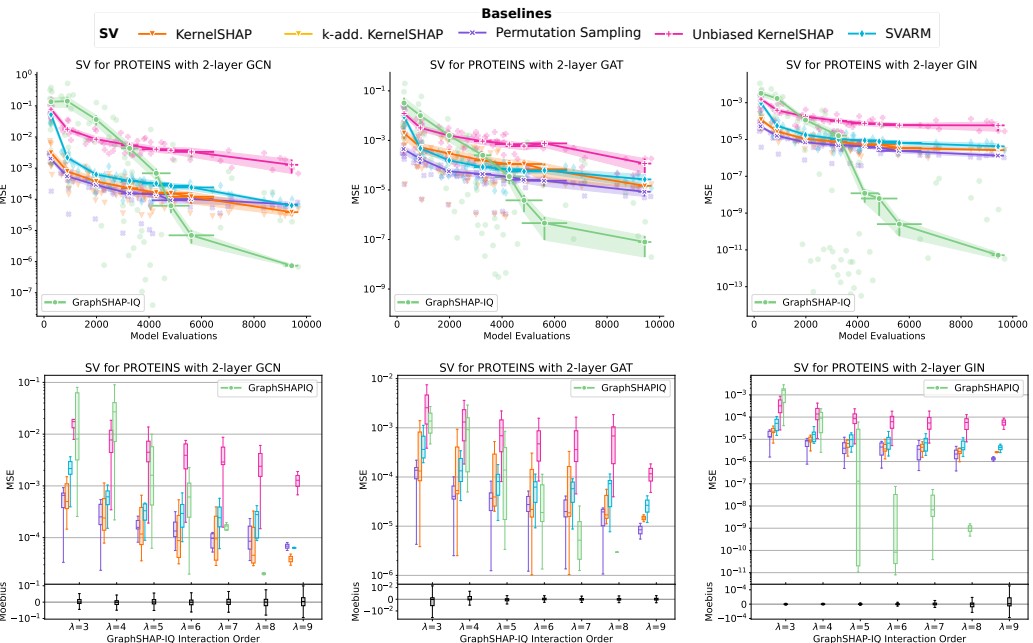

Figure 15: Comparison of GraphSHAP-IQ's approximation quality with model-agnostic baselines on 10 graphs with $30 \leq n \leq 40$ nodes from the *PRT* dataset for a 2-layers GCN (left), GAT (middle) and GIN (right). The top row presents the MSE for each estimation (dots) and averaged over $\lambda$ (line) with the standard error of the mean (confidence band). The bottom row shows the same information including the MIs as box plots for each $\lambda$.

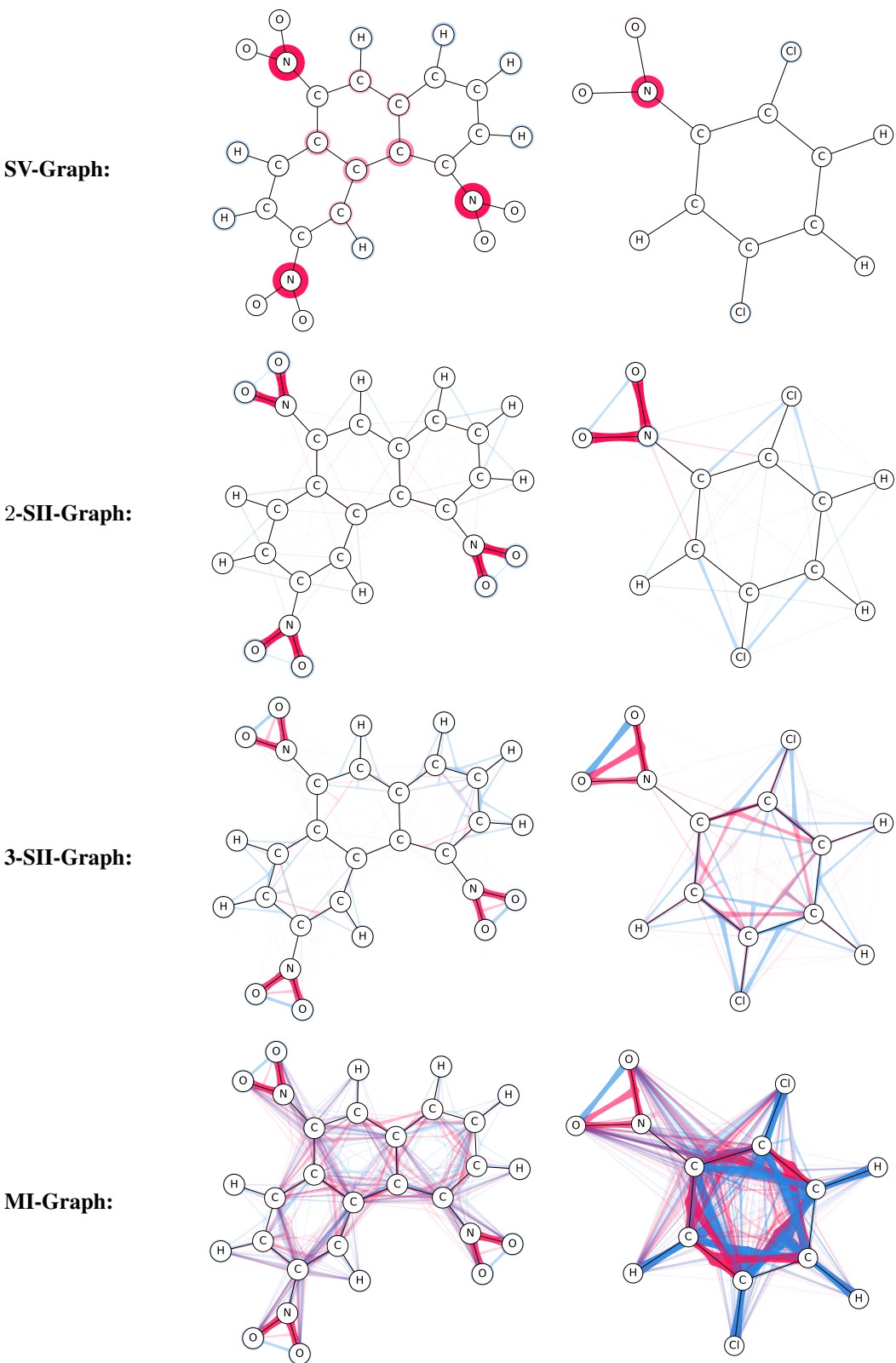

Figure 16: Additional SI-Graphs for molecule structures of the *MTG* dataset; molecule *71* with 30 atoms (left) and molecule *189* with 14 atoms (right). The model is a 2-layer GCN.

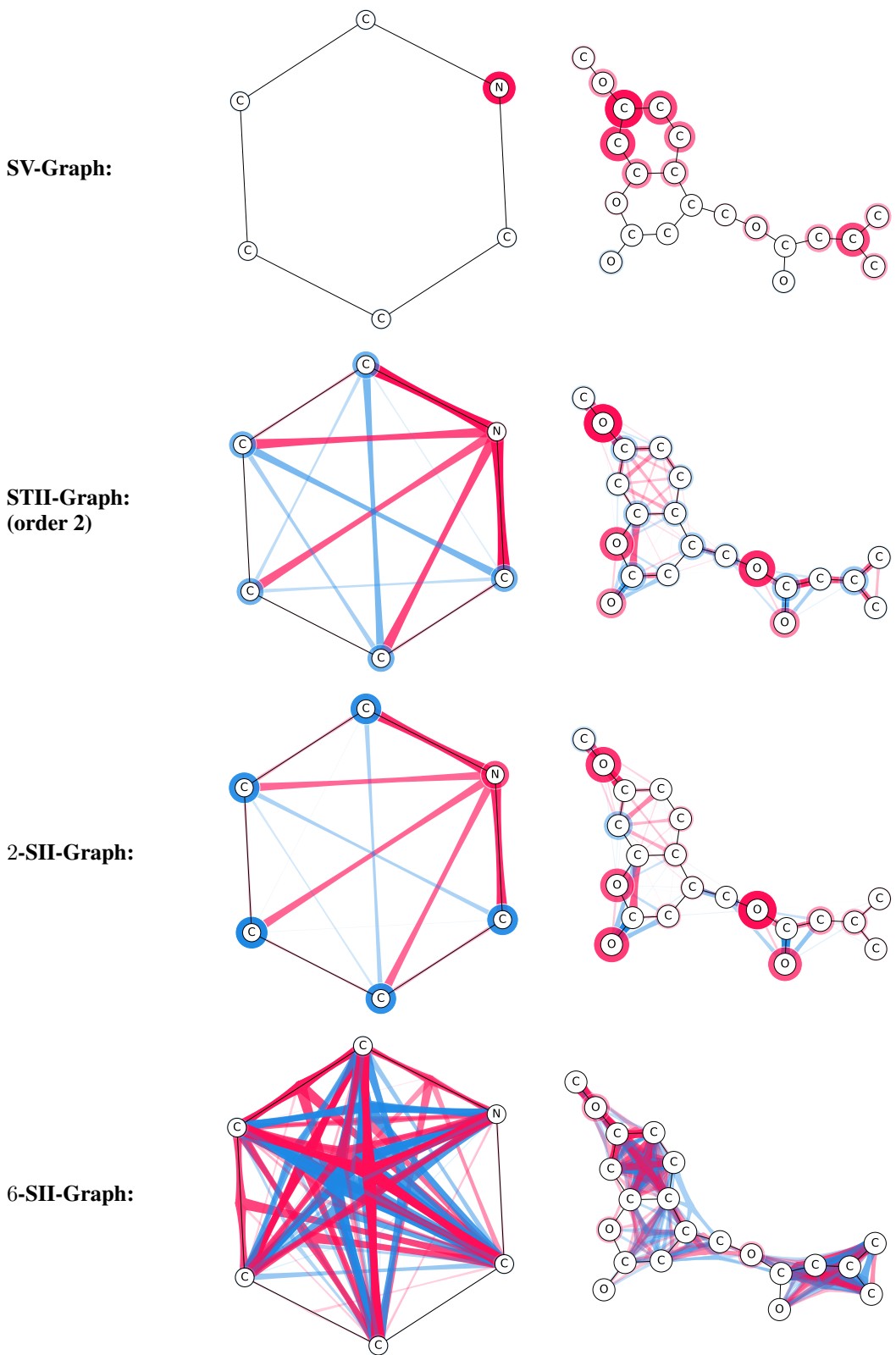

Figure 17: Additional SI-Graphs for molecule structures of the *BNZ* dataset; molecule *Pyridine* with 6 atoms (left) and molecule *57* with 21 atoms (right). The model is a 3-layer GAT.

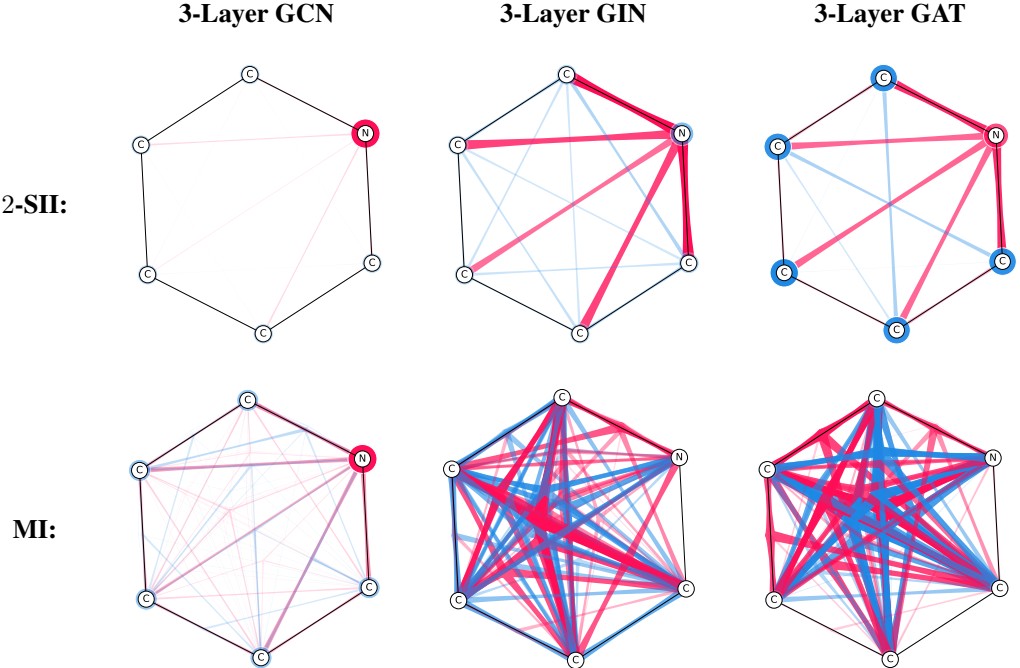

Figure 18: Comparison of the SI-Graphs for the pyridine molecule for the three model architectures fitted on *BNZ*. All models accurately predict the molecule to be non-benzene.

