# OpenReview forum: "Exact Computation of Any-Order Shapley Interactions for Graph Neural Networks"
_ICLR.cc/2025/Conference — ICLR 2025 Poster_

### Official Review · Reviewer_HTEk · 2024-10-25

**Soundness:** 3
**Presentation:** 3
**Contribution:** 3
**Rating:** 6
**Confidence:** 3

**Summary:**

This paper proposes a method for efficient calculation of exact and approximate any-order Shapley Interactions by leveraging GNN structure and node receptive fields to filter out trivial interactions, eliminating unnecessary computations and significantly accelerating processing. For highly connected graphs or very deep GNNs, the paper introduces an approximation technique to ensure computational feasibility. Experiments demonstrate substantial acceleration and low error for the approximation method.

**Strengths:**

1.	The figures are well plotted, particularly Fig. 2.
2.	This paper takes an innovative approach by leveraging the structural characteristics of GNNs to accelerate the computation of any-order Shapley Interactions, while ensuring exact results.
3.	The experiments cover a diverse range of datasets and GNN architectures, providing comprehensive qualitative and quantitative results that demonstrate the method’s efficiency and low approximation error.

**Weaknesses:**

1. The restriction to a linear readout function may limit the method’s broader applicability.
2.	Higher-order interactions could make the interpretations for the visualization more challenging for users.
3.	The extensive use of varied notations can be difficult to follow without a notation table.

**Questions:**

1. In Fig.4 right, it seems that we can just plot the top-k most important 2-node group to remove the unimportant ones and get a clearer visualization. And the top relevant groups seem to be the same, i.e, N-O? It would be interesting to compare the top-k most important groups of the exact SHAP and approximated SHAP.
2. Accuracy and computation expense needs trade-off when using SHAP. How much faster/slower is the proposed method than other approximation methods of SHAP? A figure with computation expense as x-axis, MSE as y-axis and each method as a point would be useful for users to decide when to use which method.
3. Is the proposed method extendable to other models? E.g., for CNN, where each input pixel also has receptive fields.

---

> ### Author Response · Authors · 2024-11-19
> **Reply to Reviewer HTEk**
>
> We **gratefully thank** the anonymous reviewer for their appreciation of our contribution of exact Shapley interactions for GNNs, and the time invested and thoughtful comments to help us  convey the contribution of our manuscript! We address the weaknesses and questions in the following:
> ### **Weaknesses**
> - **W1 (linear readout)**: Yes, we discuss this limitation in Section 5 and 6. In this case, it will arguably be infeasible to compute MIs, since they are not restricted by the graph structure within the GNN, as shown in Figure 6. However, our empirical results indicate that SIs are quite similar to GNNs with non-linear readouts. In our view, it is thus not advisable to use the paradigm of GraphSHAP-IQ, i.e. exact computation of MIs and deriving SIs from them, but rather rely on approximation methods of SIs directly. Overall, we strongly believe that our theoretical results and the introduction of MIs to understand interactions in GNNs will enable the development of approximation methods specifically tailored to GNNs, since our results still hold on intermediate layers of the GNN, e.g. before non-linear readout. In this context, we have already extended GraphSHAP-IQ with such a graph-inspired approximation variant. However, our main contribution in this paper is the exact computation and theoretical results on sparse MIs.
> - **W2 (visualization)**: Thank you for raising this important point! We fully agree that higher-order visualizations are more challenging to interpret. The SIs yield a flexible trade-off between complexity (of visualization) and faithfulness (to the game). For standard two-way interactions, we rely on a modified network plot, which is standard for graphs [1]. Moreover, with the SI-Graph (Definition 3.1 and e.g. Figure 1), we propose an intuitive visualization technique that extends this concept to **any-order interactions**. Yet, exploring other visualization and human-centered post-processing of SIs remains an important direction for future research.
> - **W3 (notations)**: Thank you for this valuable suggestion! In the revised version, we added a notation table in the appendix.
> ### **Questions**
> - **Q1 (top-k)**: Yes, in this particular example, the approximated TOP-2 and TOP-6 interactions fully coincide with the ground-truth TOP-2 and TOP-6 set (independent of order). In practice, the choice of $k$ is however very critical and a good choice for $k$ is unknown. To illustrate this, we conducted a small study on this instance by measuring the ratio of approximated TOP-k interactions with the set of TOP-k ground-truth interactions (independent of their order) of varying $k$, which we collect in the following table for the discussed example (row 1), averaged over 100 graphs from MTG with 20-40 nodes for SV (row 2), 2-SII (row 3), 3-SII (row 4). The results show that TOP-k approximated interactions generally do not agree with TOP-k ground-truth interactions, which gets substantially worse for higher orders.
> |k|1|2|3|4|5|6|7|8|9|10|
> |-|-|-|-|-|-|-|-|-|-|-|
> |**Example Figure 1/4**|0|**1**|0.67|0.75|0.8|**1**|0.86|0.88|0.8|0.81|
> |**SV**|0.69|0.77|0.78|0.82|0.82|0.82|0.82|0.83|0.85|0.86|
> |**2-SII**|0.62|0.76|0.82|0.84|0.81|0.80|0.80|0.81|0.82|0.83|
> |**3-SII**|0.45|0.51|0.49|0.50|0.48|0.48|0.47|0.48|0.49|0.49|
>
> - **Q2 (runtime)**: Thank you for highlighting the computational aspects! To clarify this aspect, we extended our experiments with an analysis of the computational complexity (see general statement above and Appendix G.2). Following your suggestion, we exchanged the middle plot of Figure 4 with the runtime analysis for GraphSHAP-IQ and the baselines. We display average MSE against runtime in log-seconds for each method, which clearly shows how baseline methods behave in terms of computational cost and performance. Notably, the runtime of GraphSHAP-IQ is similar to the baseline methods for order 1 (SV). In contrast to the baselines, the runtime is unaffected by increasing explanation orders, while still providing exact explanations.
>
> - **Q3 (CNNs)**: Thank you for this brilliant remark! Yes, our theoretical results apply to CNNs, provided that there is a linear pooling and linear readout after the convolutions. However, for CNNs these assumptions are less common than for GNNs. Yet, our theoretical results apply to any model with spatially restricted features, e.g. topological deep learning [2], as long as these features are only linearly transformed for prediction. We added this remark to future work.
>
> [1] [Inglis, Alan, Andrew Parnell, and Catherine B. Hurley. "Visualizing variable importance and variable interaction effects in machine learning models." _Journal of Computational and Graphical Statistics_ 31.3 (2022): 766-778](https://www.tandfonline.com/doi/full/10.1080/10618600.2021.2007935)
>
> [2] [Papillon, Mathilde, et al. "Architectures of Topological Deep Learning: A Survey of Message-Passing Topological Neural Networks."  (2023)](https://arxiv.org/abs/2304.10031)

---

> > ### Comment · Reviewer_HTEk · 2024-11-25
> >
> > I appreciate the detailed response and additional experiment. I will keep my score.

---

### Official Review · Reviewer_cY3Q · 2024-10-30

**Soundness:** 1
**Presentation:** 1
**Contribution:** 1
**Rating:** 3
**Confidence:** 2

**Summary:**

The paper studies the interpretability of graph neural networks (GNNs) via Shapley Interactions (SIs). Specifically, it explores quantifying node contributions by computing exact SIs. The paper proposes an any-order SIs computation method named GraphSHAP-IQ, which can significantly reduce the complexity of exact SIs computation. Finally, it conducts extensive experiments to validate the effectiveness of GraphSHAP-IQ and complexity reduction.

**Strengths:**

- The figures on the paper are well-constructed and clearly convey the intended information.

**Weaknesses:**

- The writing in the paper needs significant improvement, as it currently makes it difficult for readers to follow the arguments and content. The issues with the writing can be summarized as follows: (1) The overall logic and flow of the paper are unclear, which hinders comprehension. (2)Several grammatical errors detract from the clarity and professionalism of the manuscript.

- The motivation for the study is not clearly articulated and does not come across as compelling. This appears to be a result of suboptimal writing throughout the paper.

- The review of related work appears to be somewhat disorganized, and it would be beneficial to provide a more detailed comparison with similar methods, such as TreeSHAP.

- The experiments provided do not convincingly demonstrate the effectiveness of the method in reducing complexity. Additional or more targeted experiments may be needed to better support this claim.

I recommend thoroughly revising the paper, enhancing the logical structure, and addressing the grammatical issues. This will greatly improve the readability and overall quality of the work.

**Questions:**

What is the purpose of showing the performance of GNN vanilla in Table 1?

---

> ### Author Response · Authors · 2024-11-19
> **Reply to Reviewer cY3Q**
>
> Dear reviewer, we are very disappointed by what you have presented as a "review" and feel that the enormous amount of work we have put into our paper has not been valued. Your feedback is completely generic, lacking in substance and providing little to no actionable points for improving the manuscript. Frankly, we don't see how we can respond to your comments in any meaningful way.
>
> To answer your question: At the core of our theoretical results is Assumption 3.4 on the GNN architecture. With the performances in Table 1 we want to show that GNNs under this assumption still achieve performances comparable to other literature.

---

> ### Comment · Area_Chair_zvPV · 2024-11-19
> **Please give the Authors more to go on!**
>
> Dear Reviewer cY3Q,
>
> I largely concur with the Authors' analysis of your review; it is a negative-leaning review that gives almost nothing concretely actionable (beyond amending and concretising relations to related work) that the Authors can do to improve their standing.
>
> If you were an Author, you probably would not appreciate receiving reviews like this.
>
> I kindly ask you to concretely specify which actions you'd like the Authors to do in order for you to consider increasing your score. For example:
>
> * What needs to be changed about the logical structure?
> * What are the key grammatical issues you have noticed?
> * What is unconvincing about the motivation of the paper?
> * Which experiments should the Authors attempt to run?
>
> If you do not provide such actions in time for the Authors to respond to them, I would likely discard your review from consideration.
>
> Best,
> AC

---

### Official Review · Reviewer_dedn · 2024-11-04

**Soundness:** 3
**Presentation:** 3
**Contribution:** 2
**Rating:** 6
**Confidence:** 3

**Summary:**

This paper identifies an invariance property in node games on graphs and demonstrates that the exponential complexity of Shapley Interactions depends only on the receptive fields of graph neural networks. Leveraging this insight, the authors propose GraphSHAP-IQ, a method for efficiently computing any-order Shapley Interactions for graph neural networks. They also introduce an approximate version of GraphSHAP-IQ, which restricts computation to the highest order of Möbius Interactions. Finally, the authors propose a visualization technique for Shapley Interactions using SI-Graph and validate their approach through experiments on various real-world applications.

**Strengths:**

- The proposed GraphSHAP-IQ method demonstrates high efficiency.
- The authors provide theoretical guarantees for GraphSHAP-IQ's computational complexity.
- Extensive experiments on real-world applications are conducted, with results clearly illustrated. Notably, the introduction of the WAQ dataset adds valuable tools for evaluating explanation methods on graphs.
- The paper is well-written and easy to follow.

**Weaknesses:**

- **Novelty**: The primary contribution of this paper lies in reducing the computational complexity of Shapley Interactions through node game invariance, limiting the calculation of Shapley Interactions within the receptive field of the graph neural network. However, this approach is not entirely novel, as it was previously proposed in other works. For example, Section 5.4 of [1] states:

  > Indeed, for a GNN model with $k$ layers, only $k$-hop neighbors of $v$ can influence the prediction for $v$, and thus receive a non-zero Shapley value. All others are allocated a null importance according to the dummy axiom and can therefore be discarded.

  Extending this approach from model-agnostic to structure-aware approximation may offer limited novelty on its own.

- **Experiments**: In Figure 4, the authors claim that GraphSHAP-IQ achieves better approximation quality than other methods, by comparing their MSE **at the same number of model evaluations**. However, as noted in the previous point, GraphSHAP-IQ’s performance advantage could be attributed simply to disregarding nodes outside the GNN’s receptive field, thereby requiring fewer model evaluations. Thus, the assertion that GraphSHAP-IQ provides superior approximation quality is unconvincing. A more balanced evaluation would involve applying the same efficiency optimization across all methods and comparing results to see if GraphSHAP-IQ still outperforms.

- **Minor Issues**: The vertical spacing between paragraphs appears missing. Additionally, some capitalized terms (e.g., SV, SI, MI) and the term “BShap” contain hyperlinks that link incorrectly to the first page of the paper.

[1] Duval, A., & Malliaros, F. D. (2021). GraphSVX: Shapley Value Explanations for Graph Neural Networks. In *Machine Learning and Knowledge Discovery in Databases. Research Track: European Conference, ECML PKDD 2021, Bilbao, Spain, September 13–17, 2021, Proceedings, Part II 21* (pp. 302-318). Springer International Publishing.

**Questions:**

See weaknesses.

---

> ### Author Response · Authors · 2024-11-19
> **Reply to Reviewer dedn**
>
> We **gratefully thank** the anonymous reviewer for their time and critical view of our work! We **hope to clarify the raised weaknesses and questions** in the following:
>
> ### **Novelty**:
> Thank you for raising this important point! For further clarification, we added a detailed discussion of related work, including GraphSVX, in the appendix (see general statement above). In fact GraphSVX (Duval & Malliaros, 2021) consider node prediction and discard (dummy) nodes outside of the receptive fields of the node embeddings. Instead, GraphSHAP-IQ considers graph prediction, and discards (dummy) interactions, that are not fully contained in any of the receptive fields.
> Notably, GraphSVX's reasoning does not apply to graph classification (there are no dummy nodes on graph level!), and none of our theoretical results can be established with their arguments. In fact, for graph classification, GraphSVX considers all nodes, and therefore proposes a model-agnostic sampling-based approximation for the SV (KernelSHAP baseline). GraphSVX for graph prediction is therefore **not structure-aware**, as mentioned in our introduction (line 078-079) and in their paper:
> >  We simply look at $f(X, A) \in R$ instead of $f_v(X, A)$, derive explanations for all nodes
> or all features (not both) by considering features across the whole dataset instead of features of v, like our global extension.
>
> In detail, GraphSVX for node classification relies on the SV and the dummy axiom. It is argued that the node embedding of a node $v$ is not affected by a node $q$ outside the $\ell$-hop neighborhood of $v$, and thus by the dummy axiom the SV of node $q$ must be zero. Their reasoning also follows formally from our Lemma 3.5, since the SV of node $q$ is constructed from the MIs of sets $S$ that contain $i$, which are never fully contained in the $\ell$-hop neighborhood and are thus zero. However, as also observed by Duval & Malliaros (2021), the argument via SV and dummy axiom **does not hold on graph level**, since there are no dummy nodes in graph classification (all nodes affect the prediction on graph level)! In contrast, we proposed to investigate the purified interactions (MIs), where we established (Proposition 3.6) that indeed on graph level **dummy interactions** actually exist (provided linear global pooling and readout). In contrast, for graph classification GraphSVX approximates the SV directly for a game with all nodes $N$ requiring $\vert \mathcal P(N) \vert = 2^n$ calls for exact computation, whereas GraphSHAP-IQ computes exact MIs for all sets in $\mathcal I = \bigcup_{i \in N} \mathcal P(\mathcal N^{(\ell)}_i)$, which requires substantially less model calls $\vert I \vert \ll 2^n$. This result is the core of our main contribution (Theorem 3.7), which allows for the efficient computation of MIs. Consequently we are able to derive exact SVs and SIs from the exact MIs, whereas GraphSVX (for graph classification) is a model-agnostic approximation.
>
> ### **Experiments**:
> Your suggestion is a great motivation for the development of new algorithms in future work, which we briefly mentioned in line 524-526! Unfortunately, our results are **not applicable to any of the baseline** methods. In short, model-agnostic approximations rely on game evaluations (masked predictions $\nu_g(T)$ that are not sparse), whereas GraphSHAP-IQ and our theoretical results rely on (sparse) MIs ($m_g(S)$). More concretely, GraphSHAP-IQ does not disregard any nodes, instead, it disregards **dummy interactions** (MIs for set of nodes) to compute all non-trivial MIs ($m_g(S)$, Proposition 3.6). From the exact non-trivial MIs (defined for the whole powerset $\mathcal P(N)$), we were able to derive the exact SVs (defined for single nodes, $\Phi_1(i)$ ) and SI (defined for sets up to size $k$, $\Phi_k(S)$). In contrast, all baseline methods directly compute SVs or SIs by Monte Carlo approximation using randomly sampled game values (masked predictions $\nu_g(T)$). Unfortunately, a zero value of the MI $m_g(S)$ does not translate to a zero-valued masked prediction $\nu_g(S)$, and thus does not allow to restrict the sampling space for the baselines. Yet, as mentioned in future work (line 524-526), we believe that our findings of dummy interactions (Proposition 3.6) could still be used to inspire novel graph-informed approximation techniques. In this context, we have already extended GraphSHAP-IQ with such a graph-inspired approximation variant, which we used for the extreme cases, where the exact computation is infeasible, cf. empirical evaluation in Experiment 4.2.  However, our main contribution in this paper is the exact computation and theoretical results on dummy interactions.

---

> > ### Comment · Reviewer_dedn · 2024-11-20
> > **Response to Authors**
> >
> > Thank you for the detailed response. I appreciate the additional comparisons to related work provided in Appendix C.1, as well as the additional experimental results in Figure 4 and Appendix G.2. I agree with the authors that GraphSHAP-IQ accounts for dummy interactions instead of dummy nodes, enabling it to work effectively on graphs. However, I still have some concerns regarding the experimental results presented in Figure 4. While the curves in Figure 4 suggest a significant improvement in the approximation quality of GraphSHAP-IQ, all the experiments were conducted under the same computational budget. This raises the possibility that the observed improvement stems primarily from the efficiency of GraphSHAP in disregarding trivial node-game interactions, and I feel Sections 4.1 and 4.2 are essentially saying the same thing. Consequently, I feel that the huge approximation quality gains shown in Figure 4 may be somewhat misleading, as other methods might achieve comparable or superior performance if they employed similar efficiency improvements. I am curious whether other methods could attain similar approximation quality given sufficient computational resources, and whether GraphSHAP-IQ continues to outperform them without leveraging the dummy interaction enhancement. That said, I am otherwise satisfied with the authors' explanations and am happy to raise my score.

---

> > > ### Author Response · Authors · 2024-11-21
> > > **Follow-up on your Response (added interaction-informed baselines)**
> > >
> > > We gratefully thank the reviewer for their quick and intriguing response. As mentioned in our previous response, applying our results directly to the baseline methods is a bit tricky, since the baselines do not use MIs, but rather game evaluations $\nu_g$ (where none can be discarded). As a consequence, the baseline methods still require exponentially many model calls for exact computation. However, as a first step, **we proposed several interaction-informed baseline methods following your suggestion**, which improves the approximation quality and runtime. We hope that the following response clarifies your concerns.
> > > - **Interaction-informed baselines**: We confirm your intuition. For higher-order explanations (order > 1) the sparsity of MIs (Theorem 3.7) indeed implies sparsity of SIs. In fact, SIs of subsets that are not contained in $\mathcal I$ (set of non-trivial MIs) are necessarily zero, since all higher-order MIs of their supersets are zero (due to the structure of $\mathcal I$), and SIs are a weighted average of these MIs. Consequently, we modified all baseline methods to ensure that these SIs are estimated with zero. We added details of this reasoning and the modification of each baseline method to a new Appendix D.3, including a brief summary in Section 3.3. We further added these interaction-informed variants in Experiment 4.2. Our results show that the interaction-informed variants substantially improve all baselines (except permutation sampling) with regard to **approximation quality and runtime**. Moreover, the observed strong differences in the example in Figure 4, right, are also eliminated for the interaction-informed variant. However, also note that there is no improvement for SVs, and interaction-informed variants still only converge to exact SIs, if all exponentially many model calls are available. This is in contrast to the GraphSHAP-IQ approximation, which yields exact values for $\lambda = n^{(\ell)}_{\max}$ requiring the optimal budget. However, as discussed in our experiments GraphSHAP-IQ’s approximation shows mixed results when higher-order interactions dominate and the interaction-informed baselines might be preferable.
> > > - **Differences between Experiments 4.1 and 4.2** Experiment 4.1 considers the complexity of **exact** SI, while 4.2 considers restricted settings and the **approximation** of SIs. In 4.1, we empirically confirm that GraphSHAP-IQ yields a substantial reduction across the benchmark datasets, and confirm that complexity scales linearly with the graph size even across instances. In 4.2 we select a few instances, and evaluate runtime and MSE for approximation between GraphSHAP-IQ, the interaction-informed baselines, and the model-agnostic baselines. We added a few lines for clarification.

---

> > > > ### Comment · Reviewer_dedn · 2024-11-22
> > > > **Response to Authors**
> > > >
> > > > Thank you for your response and for including the interaction-informed baselines. The newly added results are promising and certainly improve the quality of the paper. I appreciate the authors' efforts in addressing my questions and providing thorough rebuttals. My main concerns have been resolved, and I have adjusted my score accordingly.

---

> > > > > ### Author Response · Authors · 2024-11-22
> > > > > **Thank you!**
> > > > >
> > > > > Thank you again for your valuable time and **constructive discussion**. We are delighted that your concerns have been resolved, and **greatly appreciate the increase in score!**

---

### Official Review · Reviewer_zYPu · 2024-11-06

**Soundness:** 3
**Presentation:** 3
**Contribution:** 3
**Rating:** 6
**Confidence:** 2

**Summary:**

The paper introduces GraphSHAP-IQ, an approach to compute any-order Shapley Interactions exactly. The authors focus on explanations for fro graph classification task. First thing, they introduced GNN-induced Graph and Node Game, they show the invariance of the node game with respect of masking outside its $\ell$ neighbourhood, where $\ell$ is the number of layers of the GNN.
Exploiting this they also show that for GNN the complexity of MIs depends only linearly in the saize of the graph and exponentialy in the connectivity of the graph. Finally experiments on real world dataset are reported.

**Strengths:**

-  The mehtod introduced in the paper is novel.
-  The method is sound  and  the authors provide robust theoretical results.
- The authors validate their approach with experiments on diverse datasets, including real-world datasets.

**Weaknesses:**

- Adding information on the algorithm's running time across different datasets and compare it with the running time of the baselines would provide more information about the applicability of the method.
- The method's efficiency heavily depends on graph sparsity and the size of receptive fields. For very dense or large graphs, the complexity may still be prohibitive.
- The algorithm assumes linear global pooling and output layers, which limits its direct application to non-linear readouts.

**Questions:**

- The paper addresses the problem for graph classification. Could this approach be extended to node classification?
- Could we use a different baseline choise intead of the mean.  Such as a random baseline and a learned baseline ?

Typo:
- Line 421 "ground truth " shold be "Ground truth".

---

> ### Author Response · Authors · 2024-11-19
> **Reply to Reviewer zYPu**
>
> We **gratefully thank** the anonymous reviewer for appreciating the novelty and theoretical results of our work. We would like to engage with you with the following point-by-point response to your questions and concerns, and **hope to make you more convinced about our contribution**.
>
> - **W1:** Thank you for this valuable suggestion. In our work, instead of measuring the actual computational time, we relied on the number of model calls as the main driver for computational complexity. This is standard in related work of the model-agnostic baseline methods, which scale linearly with number of model calls. To verify this behavior, we added a runtime analysis (see general statement for details).
> - **W2**: We fully agree that exact computation depends on the receptive fields and graph density (sparsity of edges), but we disagree that larger graphs are a problem! As we show theoretically (Theorem 3.7) and empirically (Experiment 4.1) the complexity does depend **at most linearly on the size** of the graph. This is in stark contrast to the model-agnostic computation, which scales exponentially with the number of nodes. While GNNs each convolutional layer increases the initial budget, the size of the graph is not a limiting factor in general, e.g. in Figure 3, left, we observe that for 2-Layer GNNs a budget of $10k$ suffices to compute exact SIs for graphs up to size 55, which would otherwise require $2^{55} \approx 10^{16}$ model calls. For the 3-Layer GNN, we require in this case between $10^7-10^8$ model calls with GraphSHAP-IQ for all instances, independent of size. To verify that the complexity scales only linearly with size, we additionally computed the $R^2$ of all fitted logarithmic curves (solid lines) in Figure 3 and other benchmark datasets in Figure 7-9, and added them to the labels. The logarithmic fit in the exponentially scaled plot (i.e. linear fit) exhibits a moderate ($R^2\approx 0.5$) to strong ($R^2>0.9$) $R^2$ fit, which validates that the complexity of the computation grows linearly with the size of the graph. Note that this behavior is **across all instances**, independent of a specific graph structure. Yet, as mentioned in the paper, in extreme cases exact computation might still be restricted, where the approximation of GraphSHAP-IQ should be used, which allows for a flexible budget range.
> - **W3**: Yes, we discuss this limitation in Section 5 and 6. In this case, it will arguably be infeasible to compute MIs, since they are not restricted by the graph structure within the GNN, as shown in Figure 6. However, our empirical results indicate that SIs are quite similar to GNNs with non-linear readouts. In our view, it is thus not advisable to use the paradigm of GraphSHAP-IQ, i.e. exact computation of MIs and deriving SIs from them, but rather rely on approximation methods of SIs directly. Overall, we strongly believe that our theoretical results and the introduction of MIs to understand interactions in GNNs will enable the development of approximation methods specifically tailored to GNNs, since our results still hold on intermediate layers of the GNN, e.g. before non-linear readout. In this context, we have already extended GraphSHAP-IQ with such a graph-inspired approximation variant. However, our main contribution in this paper is the exact computation and theoretical results on sparse MIs.
> - **Q1**: Yes, of course! All our results directly transfer to node prediction, which relates to the simplistic setting of a single node game $\nu_i$ from Definition 3.2. Note that this is the trivial setting, where Assumption 3.4 (linear pooling and readout) is not even required! We can directly compute all MIs and corresponding SVs or SIs using Theorem 3.3. However, in this simplistic setting, arguments from GraphSVX (Duval & Malliaros, 2021) could be applied, which remove nodes that do not affect the node embedding using the dummy axiom. At the core of our contribution are graph predictions, and specifically the usage of MIs to derive efficient computation on the graph level. As noted by Duval & Malliaros (2021) their reasoning via SVs does not transfer to graph prediction, because there are no dummy features on the graph level. However, as established in our work (Proposition 3.6), there are indeed **dummy interactions** (MIs), which allow to efficiently compute MIs on graph level and then derive SVs and SIs from later on.
> - **Q2**: Thank you for this question! We discuss this in the paragraph "Node Masking" and now clarify this in lines 284-286. Our theoretical results hold for **any masking technique**, provided that it can be applied to any subset of nodes. We decided to rely on BSHAP as a generally applicable, well-established and theoretically well-understood choice, and it remains important future work to explore other choices that are more suitable to specific use cases.

---

> > ### Comment · Reviewer_zYPu · 2024-11-26
> > **Thanks the authors for their replies, keeping my score**
> >
> > Sorry, about W2 I meant it would be proibitive for large highly connected graphs. Theorem 3.7  still has exponential dependence on the size of the largest l-hop neighborhood.
> > However, I'm satisfied with authors responses and explanations. I'll maintain my current score.

---

### Author Response · Authors · 2024-11-19
**General Statement and Runtime Analysis**

We **gratefully thank** the anonymous reviewers **zYPu**, **dedn** and **HTEk** for their time invested in reviewing our manuscript, and their valuable suggestions and discussions.
With our revision, we introduced four $\color{blue}changes$ in the manuscript:
## Improvements and Minor Changes
1. Conducted and added a runtime analysis to confirm that the main driver of GraphSHAP-IQ's runtime are indeed the number of model calls in Appendix G.2 (requested by reviewers **zYPu**, **dedn** and **HTEk**)
2. We added a notation table in Appendix A (requested by reviewer **HTEk**)
3. We added extended related work in Appendix C.1 with a detailed comparison of other approaches, such as GraphSVX (requested by reviewer **dedn**)
4. Removed hyperlinks from all acronyms (requested by reviewer **dedn**)

## Runtime Analysis (requested by reviewers zYPu, dedn, and HTEk)
A mutual concern raised was our analysis of computational efficiency of GraphSHAP-IQ with the model-agnostic sampling-based approximation baselines. In our comparison, we relied on the number of model calls as the main driver of computational complexity and ensured that GraphSHAP-IQ and all baselines were given the same budget of model calls. This is standard in the approximation literature, and it was confirmed that the runtime of baselines scales linearly with number of model calls, e.g. cf. [Tsai et al. (2023)](https://www.jmlr.org/papers/v24/22-0202.html) or Appendix D.1 in [Fumagalli et al. (2024)](https://proceedings.mlr.press/v235/fumagalli24a.html). To empirically verify that the number of model calls is also the main driver in GraphSHAP-IQ, we conducted a runtime analysis. For a 2-Layer GCN and the MTG dataset, we selected 100 graphs with 20-40 nodes that require less than 10k model calls for exact computation with GraphSHAP-IQ (baselines require 1m-1b). Similar to our experiments, all baseline methods were given the same budget as GraphSHAP-IQ. The results of this runtime analysis were added to Appendix G.2. In Figure 10 (upper row), we plotted the runtime of all methods for all these instances against the number of model calls, which results in an almost perfect linear fit for GraphSHAP-IQ ($R^2 > 0.97$). Moreover, the size of the graphs does barely affect the runtime (lower row). With increasing interaction order, most baselines (KernelSHAP-IQ, SVARM-IQ, SHAP-IQ), even require substantially more runtime given the same budget, since their number of estimated interactions increases drastically. In contrast, GraphSHAP-IQ is almost unaffected by the increasing explanation order. In summary, we empirically confirm that the number of model evaluations is the main driver of computational complexity in GraphSHAP-IQ.

---

> ### Author Response · Authors · 2024-11-21
> **Additional Changes after Discussion with Area Chair zvPV and Reviewer dedn**
>
> After discussion with the area chair **zvPV** and follow-up response of reviewer **dedn**, we $\color{blue}\text{added}$ the following:
> - We restructured the related work section in the main paper, and moved the comprehensive overview to Appendix C. Moreover, we added a detailed comparison with other approaches, such as GraphSVX or TreeSHAP in Appendix C.1. (requested by reviewer **dedn**, **cY3Q** and area chair **zvPV**)
> - For each model-agnostic baseline for approximation of SIs, we added an interaction-informed variant. Accordingly, we adapted Section 3.3, and added a new section with details in Appendix D.3. Moreover, we extended the empirical analysis in Section 4.2. (requested by reviewer **dedn**)

---

### Meta-Review · Area_Chair_zvPV · 2024-12-21

**Metareview:**

This paper presents a very interesting contribution to explainable AI on graph structures, through a graph-oriented method for computing Shapley Values, dubbed GraphSHAP-IQ.

All in all, I find the paper to be a compelling read; beautifully written, with illustrious figures, and a clear set of useful results and pointers to relevant related work. I think the paper is in good shape to be accepted!

**Additional Comments On Reviewer Discussion:**

Initially there was no clear majority in support of accepting this paper, which was overturned by the time the Authors posted their rebuttal. Further, I have chosen to discard the review of Reviewer cY3Q, by their own confession, as this paper was not in their area of expertise and they were unable to prepare a well-targeted review. Coupled with score increases elsewhere, I consider the paper to now be unanimously supported for acceptance. While support is in the "weak" acceptance category across all Reviewers, I see no clear outstanding issues that need resolving, and I have no reservations to recommend acceptance in the paper's current form.

---

### Decision · Program_Chairs · 2025-01-22

Accept (Poster)